# A transcriptional network required for bradyzoite development in *Toxoplasma gondii* is dispensable for recrudescent disease

Sarah L. Sokol-Borrelli[1,5], Sarah M. Reilly[1,5], Michael J. Holmes [2],
Stephanie B. Orchanian [3], Mackenzie D. Massmann[1], Katherine G. Sharp [1],
Leah F. Cabo [1], Hisham S. Alrubaye[1], Bruno Martorelli Di Genova [4],
Melissa B. Lodoen[3], William J. Sullivan Jr.[2] & Jon P. Boyle [1] ✉

Identification of regulators of *Toxoplasma gondii* bradyzoite development and cyst formation is the most direct way to address the importance of parasite development in long-term persistence and reactivation of this parasite. Here we show that a *T. gondii* gene (named *Regulator of Cystogenesis 1*; *ROCY1*) is sufficient for *T. gondii* bradyzoite formation in vitro and in vivo. *ROCY1* encodes an RNA binding protein that has a preference for 3′ regulatory regions of hundreds of *T. gondii* transcripts, and its RNA-binding domains are required to mediate bradyzoite development. Female mice infected with Δ*ROCY1* parasites have reduced (>90%) cyst burden. While viable parasites can be cultivated from brain tissue for up to 6 months post-infection, chronic brain-resident Δ*ROCY1* parasites have reduced oral infectivity compared to wild type. Despite clear defects in bradyzoite formation and oral infectivity, Δ*ROCY1* parasites were able to reactivate with similar timing and magnitude as wild type parasites for up to 5 months post-infection. Therefore while *ROCY1* is a critical regulator of the bradyzoite developmental pathway, it is not required for parasite reactivation, raising new questions about the persisting life stage responsible for causing recrudescent disease.

Eukaryotic parasites are renowned for their complex, multihost life cycles that require transitions between life stages that differ dramatically in their morphology, metabolism, and reproductive niche(s). *Toxoplasma gondii* is no exception, a tissue-dwelling coccidian that has infected a third of the world's population but which has dramatic differences in disease outcome ranging from being benign to causing lethal toxoplasmic encephalitis. A remarkable feature of the *T. gondii* life cycle is the ability to persist for many years in the infected host, and this chronic phase of infection is thought to be driven primarily by the bradyzoite, a slow-growing life stage that becomes surrounded by a protein- and sugar-rich cyst wall during its development. It has long been presumed that bradyzoite development and tissue cyst formation is a response to the host immune response and is required for long-term *T. gondii* persistence[1,2]. In mice, the most well-recognized

[1]Department of Biological Sciences, Dietrich School of Arts and Sciences, University of Pittsburgh, Pittsburgh, PA, USA. [2]Department of Pharmacology & Toxicology, Indiana University School of Medicine, Indianapolis, IN 462022, USA. [3]Department of Molecular Biology & Biochemistry, University of California, Irvine, CA, USA. [4]Department of Microbiology and Molecular Genetics, University of Vermont Larner College of Medicine, Burlington, VT, USA. [5]These authors contributed equally: Sarah L. Sokol-Borrelli, Sarah M. Reilly. ✉e-mail: boylej@pitt.edu

site of *T. gondii* persistence is within the brain, where it resides within neurons and can cause lethal encephalitis when immune surveillance is altered by experimental treatment or disease[3–6]. While multiple genes have been identified that are associated with the tachyzoite to bradyzoite transition[7–10], to date, the complete regulatory networks driving this transition are unknown, as are the mechanisms that lead to their activation. A less than complete understanding of how this process is regulated transcriptionally and executed by downstream effectors presents a barrier to the development of new therapies targeting this drug-refractory life stage. Moreover to date, there is limited evidence for the importance of bradyzoite formation and the cyst wall itself in the persistence of *T. gondii* life stages that are capable of recrudescence despite their presumed (and logical) importance in this process.

As a means to identify additional regulators of cyst formation in *T. gondii*, we have taken advantage of the *T. gondii*/*H. hammondi* comparative system. *H. hammondi* is the nearest extant relative of *T. gondii* and shares nearly all of its genes across chromosomes that are almost entirely co-linear[11,12]. These two parasite species are morphologically and antigenically similar, and share the same definitive host and transition between similar life cycle stages[13–15]. Despite extensive interspecies similarities, *H. hammondi* follows an obligately heteroxenous life cycle[14–16] while *T. gondii* has a facultative heteroxenous/homoxenous life cycle. This critical difference is driven by the unique ability of *T. gondii* bradyzoites to convert back to tachyzoites in vitro and presumably in vivo and cause severe disease and damaging inflammatory responses in the host[17]. This unique ability to run its life cycle in reverse (bradyzoite back to tachyzoite) is thought to be required for *Toxoplasma* to cause severe disease in immune-compromised organ transplant patients, cancer patients, and those with HIV/AIDS[18,19]. Bradyzoite to tachyzoite interconversion also likely permits *T. gondii* transmission between intermediate hosts[20] (for example, cow or lamb to human via consumption of undercooked meat). However, to date the exact life stages responsible for the recrudescent disease are unknown, although the bradyzoite-containing tissue cysts that are characteristic of *T. gondii* and other tissue-dwelling coccidia are the most logical culprit.

*H. hammondi's* strict obligate heteroxenous life cycle restricts its ability to initiate infection in organisms other than its definitive feline host after it transitions into its bradyzoite life form[13,16,21,22]. The stark contrast in when and how bradyzoites form in each species makes interspecies comparison a promising strategy for uncovering critical components driving the bradyzoite developmental program. Moreover, *H. hammondi* cannot be induced to form bradyzoites in its early life stages when treated with alkaline pH stress[16], a potent inducer of cystogenesis in *T. gondii*[23]. Yet, *H. hammondi* predictably and reliably forms tissue cysts spontaneously, which is also accompanied by robust transcriptional changes resembling those observed during *T. gondii* bradyzoite differentiation[13,16]. Studies designed to characterize the *T. gondii* bradyzoite have led to the identification of several bradyzoite-specific proteins and have characterized the major transcriptional differences that occur during bradyzoite development in *T. gondii*[24–29]. However, until recently, the mechanisms used by *T. gondii* to initiate bradyzoite development were elusive. The first major class of transcription factors identified in Apicomplexans were AP2 transcription factors[30], which broadened our understanding of the regulation of bradyzoite development. These transcription factors play fundamental roles in regulating stage conversion-associated gene expression in closely related *Plasmodium* species[31]. In *T. gondii*, this family of proteins are important for cystogenesis and function as activators and repressors of stage conversion-associated gene expression[7–9,32]. In addition to AP2 factors, another *T. gondii* transcription factor that regulates the tachyzoite to bradyzoite transition, Bradyzoite-Formation Deficient-1 (BFD1;TGME49_200385), was identified as a transcription factor that is absolutely required for tissue cyst formation in vitro and in vivo[10]. The identification of BFD1 as a regulator of differentiation was an important leap for the field in understanding the bradyzoite developmental pathway in *T. gondii*. However, how BFD1 itself is activated, as well as its downstream targets that are most critical for bradyzoite development, remain unknown.

Here, we used an interspecies comparative approach that exploits our knowledge of the critical differences in tissue cyst formation between *T. gondii* and *H. hammondi* to identify factors that are important for driving cystogenesis. Using this system, we identified a *T. gondii* gene, *Regulator of Cystogenesis 1* (*ROCY1*; also named *BFD2* in a recently published study[33];), as a gene encoding an RNA-binding protein that interacts with the regulatory and coding regions of multiple bradyzoite-associated mRNA transcripts and is necessary and sufficient for cyst formation. We situate ROCY1 firmly within the BFD1-driven bradyzoite developmental pathway and also use ROCY1 and BFD1 mutants to directly test the importance of bradyzoite formation in vivo for the persistence of *T. gondii* stages that are capable of reactivation, a critical feature of the *T. gondii* life cycle that is responsible for severe disease in the immunocompromised.

## Results

### Comparative transcriptomics between *T. gondii* and *H. hammondi* identifies candidate genes involved in tissue cyst development

We leveraged previously characterized differences between *T. gondii* and *H. hammondi* in spontaneous and alkaline pH-induced tissue cyst formation[16] to identify new genes in the bradyzoite/cyst development pathway. Specifically, when *T. gondii* sporozoites are exposed to alkaline pH-induced stress conditions at 3 days post excystation (DPE), they robustly form *Dolichos biflorus* agglutinin (DBA)-positive tissue cysts in vitro. In contrast *H. hammondi* fails to form DBA-positive tissue cysts under these conditions, but spontaneously forms DBA-positive tissue cysts beginning at 12DPE (Summarized in Fig. 1a)[16].

We hypothesized that some of the genes with altered transcript abundance during alkaline pH stress exposure in *T. gondii* and during spontaneous development in *H. hammondi* would have critical regulatory roles in tissue cyst development. To identify these genes we performed RNA sequencing experiments on *T. gondii* and *H. hammondi* sporozoite-derived infections exposed to normal or bradyzoite-induction conditions (BIC; alkaline pH, low serum, and $CO_2$ starvation) for 48 h beginning at 3 DPE. We then integrated these transcriptional profiles with previously published data[16] to identify genes that had 1) significant changes in transcript abundance ($|Log_2FC| > 1$, $P_{adj} < 0.01$) in *T. gondii* in BIC compared to control conditions in at least 1 of 2 experiments, 2) genes that did not have significant changes in *H. hammondi* during BIC relative to control conditions in at least one of 2 experiments, and 3) genes with significant differences in transcript abundance during spontaneous tissue cyst formation in *H. hammondi* (D4 vs D15). We identified 80 candidate genes fitting these criteria (Fig. 1a, b). To prioritize these 80 candidates for further investigation, we sought to identify potential transcription factors and/or nucleic acid-binding proteins that could be responsible for driving changes required for the transition from tachyzoite to bradyzoite. To do this we first removed candidates with a predicted signal peptide or transmembrane domains, leaving 45 candidates. Next, we identified 22 genes containing predicted nuclear localization sequences using NLSmapper[34] and NLStradamus[35] using default settings. Finally, to further prioritize our candidate genes, we utilized data from a genome-wide CRISPR screen in *T. gondii* that predicts whether a gene contributes to tachyzoite fitness in vitro[36]. We used these data to remove those genes with fitness scores <−1, leaving us with 19 final high priority candidates. (Fig. 1c). The gene expression profiles of these 19 candidates are shown in Fig. 1d.

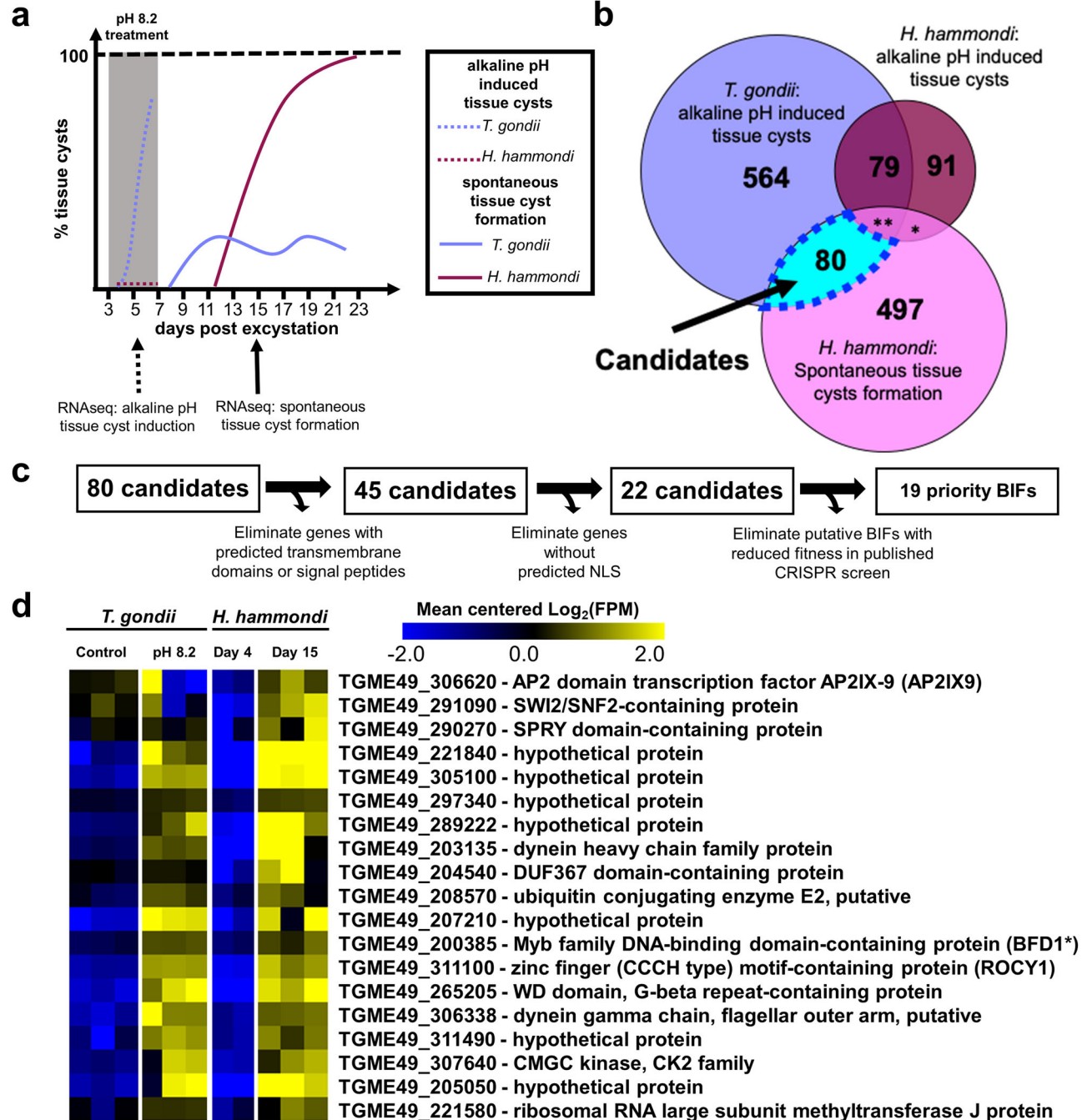

**Fig. 1 | Comparative transcriptomics between *T. gondii* and *H. hammondi* identifies candidate genes involved in tissue cyst development. a** Schematic depicting spontaneous and alkaline pH induced tissue cyst formation phenotype over a 23-day time course in *T. gondii* VEG and *H. hammondi* Amer. **b** Venn diagram showing genes with significant (|Log$_2$ Fold Change| ≥ 1, $P_{adj}$ < 0.01) differences in transcript abundance during alkaline pH induced stress in *H. hammondi* and *T. gondii* (at least 1 of 2 independent experiments) and during spontaneous development in *H. hammondi*. * = 7, ** = 22 . **c** Schematic representing the strategy used to identify priority candidates predicted to be involved in tissue cyst formation. **d** Mean-centered heat map showing 19 priority candidate BIFs. For clarity data from *T. gondii* and *H. hammondi* were mean-centered separately. All conditions had 3 independent samples except for day 4 *Hammondia hammondi*, which had *N* = 2. *BFD1 was identified and described in Waldman et al. Cell 2020. Data are provided as a Source Data File.

## Genetic ablation of TGVEG_311100 impairs cyst formation in vitro

To identify cyst-regulatory genes among these candidates we disrupted 3 of them (and *TGVEG_200385/BFD1* as a calibration control; see below) using CRISPR/Cas9 by transfecting parasites with a plasmid harboring: Cas9-GFP under the control of the *T. gondii* SAG1 promoter; a U6 gRNA expression cassette[37] that we modified to express a gene of interest-specific guide. We co-transfected parasites with a linear insertion cassette encoding either the HXGPRT selectable marker for use in CEPΔHXGRPT-GFP-LUC (TgCEP) parasites or the DHFR-TS selectable marker for use in TgVEG. After cloning and validation using PCR, we focused our efforts on ablating *TGVEG_221840*, *TGVEG_207210*, *TGVEG_311100*, and *BFD1* (Fig. S1A–F). In all cases we also transfected parasites with the CAS9 plasmid harboring an irrelevant gRNA and insertion cassette and each knockout had its own passage-matched control clone.

After 48 h in BIC, we identified no significant differences in the number of DBA-positive vacuoles between the TGVEGΔ*221840*

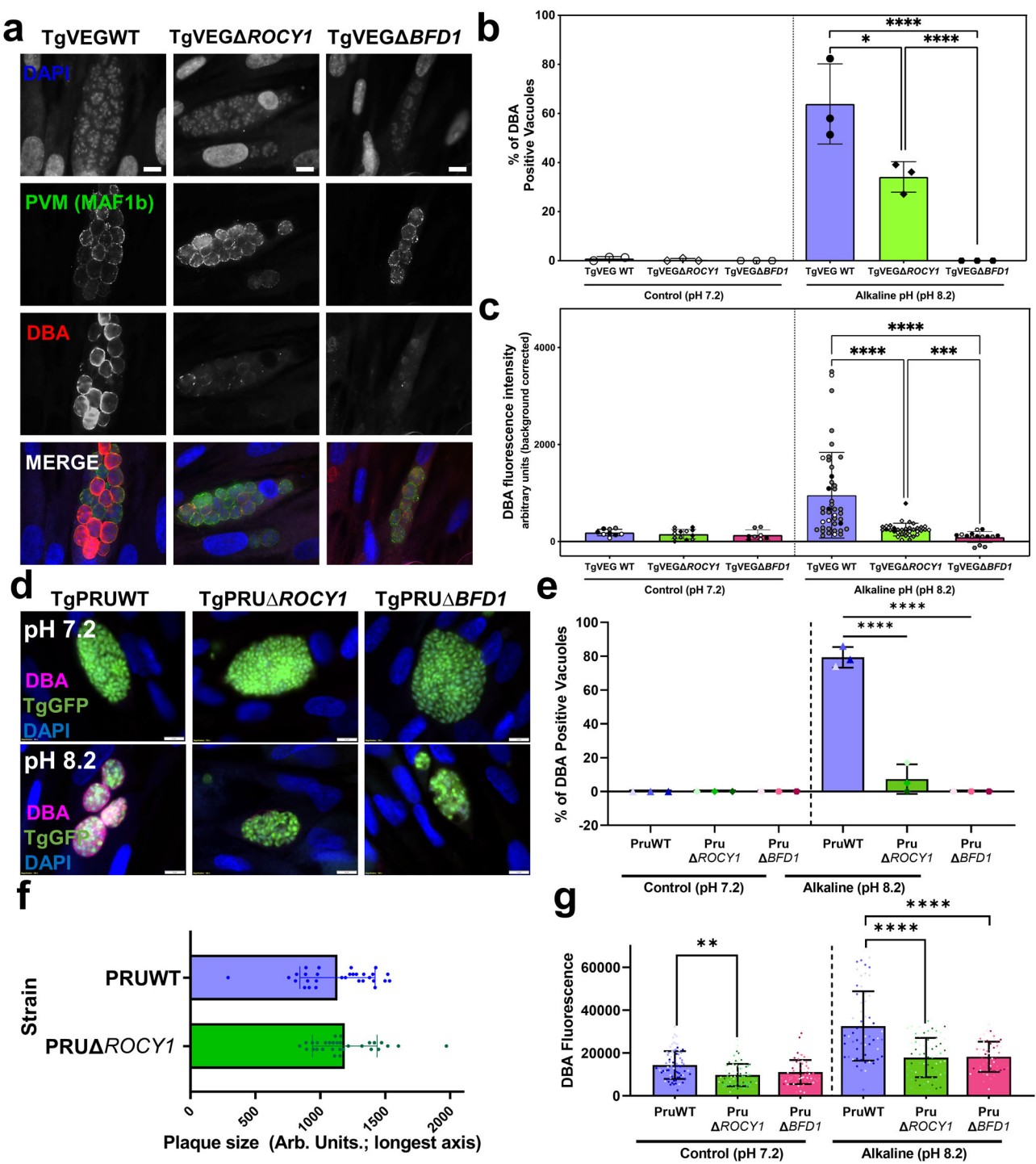

(Fig. S1G) and TGVEGΔ*207210* (Fig. S1H) knockouts relative to their passage-matched controls. However, we did observe a significant decrease (*P* = 0.0011) in the number of DBA-positive vacuoles for TgVEGΔ*311100* parasites (-25%) compared to wild type parasites (-65%; Fig. S1I). These data suggested that *TGVEG_311100* was an important gene for stress-induced cystogenesis and we named this factor Regulator of Cystogenesis 1 (ROCY1). Previous work identified BFD1 as necessary and sufficient for cyst formation in *T. gondii*[10] and given its importance in differentiation we wanted to use this a means to calibrate ROCY1 cyst defects. Therefore, we generated TgVEG strains with a disruption in the BFD1 locus (knockout validation in Fig. S1F) with the same passage history as our TgVEGΔ*ROCY1* parasite strain. As

expected both TgVEGΔROCY1 and TgVEGΔ*BFD1* parasites formed significantly fewer DBA-positive cysts (Fig. 2a, b) compared to wild type controls (*P* = 0.03 for ΔROCY1 and <0.001 for ΔBFD1). While the defect in DBA-positive vacuole formation was less robust in TgVEGΔROCY1 compared to TgVEGΔ*BFD1*, when we quantified DBA staining intensity, we found a significant difference between TgVEGΔ*ROCY1* and WT parasites and between TgVEGΔ*BFD1* and WT parasites (*P* < 0.001 for both; Fig. 2c). TgVEGΔBFD1 parasites also had significantly lower DBA staining intensity compared to TgVEGΔ*ROCY1* (*P* = 0.006; Fig. S2C). Together, these data suggest that ROCY1 plays an important role in the development of tissue cysts in vitro with a similar magnitude as BFD1.

**Fig. 2 | Genetic ablation of ROCY1 *T. gondii* VEG and PRU strains disrupts stress-induced cyst formation. a** Representative images of *T. gondii* VEG-containing vacuoles exposed to alkaline pH stress for 48 h. DNA was stained with DAPI (blue), the parasitophorous vacuole membrane (PVM) was stained with an anti-MAF1b mouse polyclonal antibody (green), and tissue cysts were stained with rhodamine-labeled *Dolichos biflorus* Agglutinin (DBA) (red). Scale bar represents 10 μm. **b** Quantification of tissue cyst formation in TgVEGΔ*ROCY1*, TgVEGΔ*BFD1*, and passaged matched TgVEG WT strains after growth in control conditions or alkaline pH. The mean and standard deviation ($N = 3$) of the percentage of DBA positive vacuoles observed in 15 parasite containing fields of view is plotted. Statistical significance was determined using ANOVA and Tukey's multiple comparisons tests on arcsine transformed data. *$p = 0.025$, ****$p < 0.0001$ . **c** Quantification of fluorescence intensity in TgVEGΔ*ROCY1* and TgVEGΔ*BFD1* from B. The mean +/- SD of the fluorescence intensity (background-corrected) is plotted. Each point represents a single vacuole. Black, gray, and white symbols indicate from which replicate.

Statistical significance was determined using a Brown-Forsythe and Welch ANOVA and Dunn's multiple comparisons (2 sided).***$p = 0.0006$ ****$p < 0.0001$.
**d** Representative images of *T. gondii* PRU-containing vacuoles exposed to alkaline pH stress for 72 h. Staining as in A. Scale bar represents 10 μm. **e** Quantification of *T. gondii* infection described in D quantifying the percentage of DBA+ vacuoles in infected coverslips ($N = 3$ per clone and condition; mean and standard deviation shown). Statistical significance was determined as in panel B and each coverslip was considered a biological replicate. **f** Mean length (in arbitrary units) +/- standard deviation of individual plaques of each strain after 7 days of growth in vitro. $N = 29$ vacuoles per parasite strain. **g** Detection and quantification of DBA fluorescent staining within all parasites (regardless of DBA-positivity; mean intensity) using KNIME software from all vacuoles categorized in e. Each data point represents and individual vacuole ($N = 78$, 75, 63, 55, 62, and 51 vacuoles per strain/condition). Statistical significance was determined as in panel (****$P < 0.0001$, **$P \leq 0.01$). Data are provided as a Source Data File.

## *ROCY1* knockouts in a type II genetic background have reduced cyst formation in vitro

Given our dependence on oocyst production in cats to produce zoites of *H. hammondi* we used a well-characterized oocyst-forming parasite strain VEG for our comparative studies of in vitro development that led to the discovery of ROCY1. However, much of the *T. gondii* cyst formation studies to date have used strains belonging to the *T. gondii* type 2 haplotype like Prugniaud (PRU)[38], which also has been modified genetically in a variety of ways to permit more sophisticated genetic manipulations. To determine the impact of ROCY1 deletion in this genetic background we used the same reagents as in TgVEG to delete *ROCY1* and *BFD1* (again for calibration) in TgPru-GFP-LUC[1]. We induced cyst formation using the same BIC as above for TgVEG and stained for DBA 4 days post-exposure. We found that PRUΔ*ROCY1* had a robust defect in DBA+ cyst formation (only ~10% of the observed vacuoles were DBA-positive cysts; Fig. 2d, e). Based on the measure of 30 7 day plaques from each strain, we detected no significant difference (Student's T-test, $P = 0.4$) in plaque size (determined by measuring the longest diameter) between wild type *T. gondii* PRU:WT and PRUΔ*ROCY1* (Fig. 2f). The defect in DBA+ cyst formation was similarly robust (at least statistically; Fig. 2e) in TgPRUΔ*BFD1*, although in contrast to the ROCY1 mutant we didn't detect any DBA+ cysts after exposure to BIC in this strain. When DBA intensity was quantified in PRUΔ*BFD1* it was also significantly lower than that in passage matched, wild type controls, but was statistically indistinguishable from PRUΔ*ROCY1* ($P > 0.05$; Fig. 2g). It is important to note here as well that even in control conditions (pH 7.2; left panel of Fig. 2g) there was detectable DBA staining throughout the parasite vacuole in both wildtype and knockout parasite lines. This staining in non-cyst forming conditions was significantly reduced in TgPRUΔ*ROCY1* compared to WT controls (Fig. 2g), and reduced, but not significantly so, in TgPRUΔ*BFD1* (Fig. 2g). These data highlight the importance of ROCY1 and BFD1 in driving bradyzoite development and DBA+ cyst formation during nutrient and pH stress, but also provide new insight into their role in driving the commonly observed phenomenon of "spontaneous" cyst formation of certain strains grown in culture[16,39].

## C-terminal tagging of ROCY1 impairs pH-induced cyst formation

As a first attempt to examine the localization of native ROCY1 we used CRISPR/CAS9 to tag the endogenous gene with a triple HA epitope in the TgPRUΔKu80ΔHXGPRT genetic background[40,41]. In PCR- and Sanger sequencing-validated tagged clones, we immunolocalized ROCY1 to the cytoplasm in both control (pH 7.2) and bradyzoite induction (pH 8.2) conditions (Fig. S2A). However when we quantified DBA+ cyst formation (indicated by DBA staining surrounding the entire circumference of the vacuole) in two endogenously tagged clones from the same transfection we found that the C-terminal tag significantly reduced BIC-driven DBA+ cyst formation (tagged strains switched at a frequency of <10% compared to over 70% in the wild type parent;

$P \leq 0.0001$ for both clones). C-terminally tagged strains also had significantly lower DBA staining intensity in BIC when quantified using immunofluorescence (Fig. S2B, C). These data suggested that C-terminally tagging ROCY1 impaired its function a it nearly phenocopied the ROCY1 knockout. We therefore used CRISPR/CAS9 to add a single FLAG tag to the N-terminus of the *ROCY1* gene (again in TgPRUΔKu80ΔHXGPRT). Using immunofluorescence we observed that qualitatively the N-terminal FLAG tagged gene product also localized to the parasite cytoplasm and could be seen in perinuclear structures similar to that observed for *T. gondii* ALBA proteins 1 and 2[42], a pair of RNA binding proteins in *T. gondii* involved in translational regulation. (Fig. S2E). Based on Western blotting where we loaded identical cell equivalents of host cell and parasite lysate under control and alkaline conditions, the FLAG-tagged protein product was readily detected in BIC at an apparent molecular weight of 130–140 KDa (Fig. S2D). This is consistent with the high transcript abundance of ROCY1 during pH-induced switching in vitro, in in vivo bradyzoites[43] and during spontaneous cyst formation in *H. hammondi* (Fig. 1d). However in contrast to the C-terminal HA tagged strain, the N-terminally FLAG-tagged clones exhibit robust DBA+ cyst formation under BIC (Fig. S2E, F), indicating that the N-terminal tagged form was functional with respect to its ability to drive cyst formation.

## ROCY1 is necessary for both induced and spontaneous cyst formation

Based on our observation that N-terminal tags are tolerated by ROCY1 while C-terminal tags are not, we placed a single HA tag immediately upstream of the predicted start codon and ROCY1 cDNA and made two complementation constructs with N-terminally-tagged ROCY1: one driving NHA-ROCY1 expression off of the putative ROCY1 promoter (Fig. 3A; encompassing 2000 bp upstream of the start codon; referred to as "Endo") and the other one driving NHA-ROCY1 using the GRA1 promoter to determine if ROCY1 (Fig. 3B). We detected ROCY1-HA-derived immunofluorescent signal in parasites complemented with the Endo-ROCY1 construct (Fig. 3C), and observed a significant increase in this ROCY-derived HA staining under BIC (Fig. 3C and quantified in Fig. 3D). Complementing TgPRUΔROCY1 with Endo-ROCY1 significantly rescued in vitro DBA+ cyst formation in BIC (Fig. 3C, E and quantified in 3F). We also isolated two clones expressing GRA1-driven NHA-ROCY1 in the ROCY1 knockout background. GRA1-driven ROCY1 complemented cyst formation in BIC for both clones (Fig. 3E and right half of Fig. 3F). However, while for one clone (clone 2 in Fig. 3F) ~40% of the vacuoles were DBA-positive in control conditions (left half of Fig. 3F; $P < 0.0001$ compared to PRUΔROCY1), clone 4 did not any DBA+ vacuoles in control conditions (Fig. 3F; left half). We repeated this with another 6 clones from separate transfections and found that 4 out of 5 GRA1-ROCY1-expressing clones made DBA+ vacuoles in control conditions during early passages (1–3), but lost this

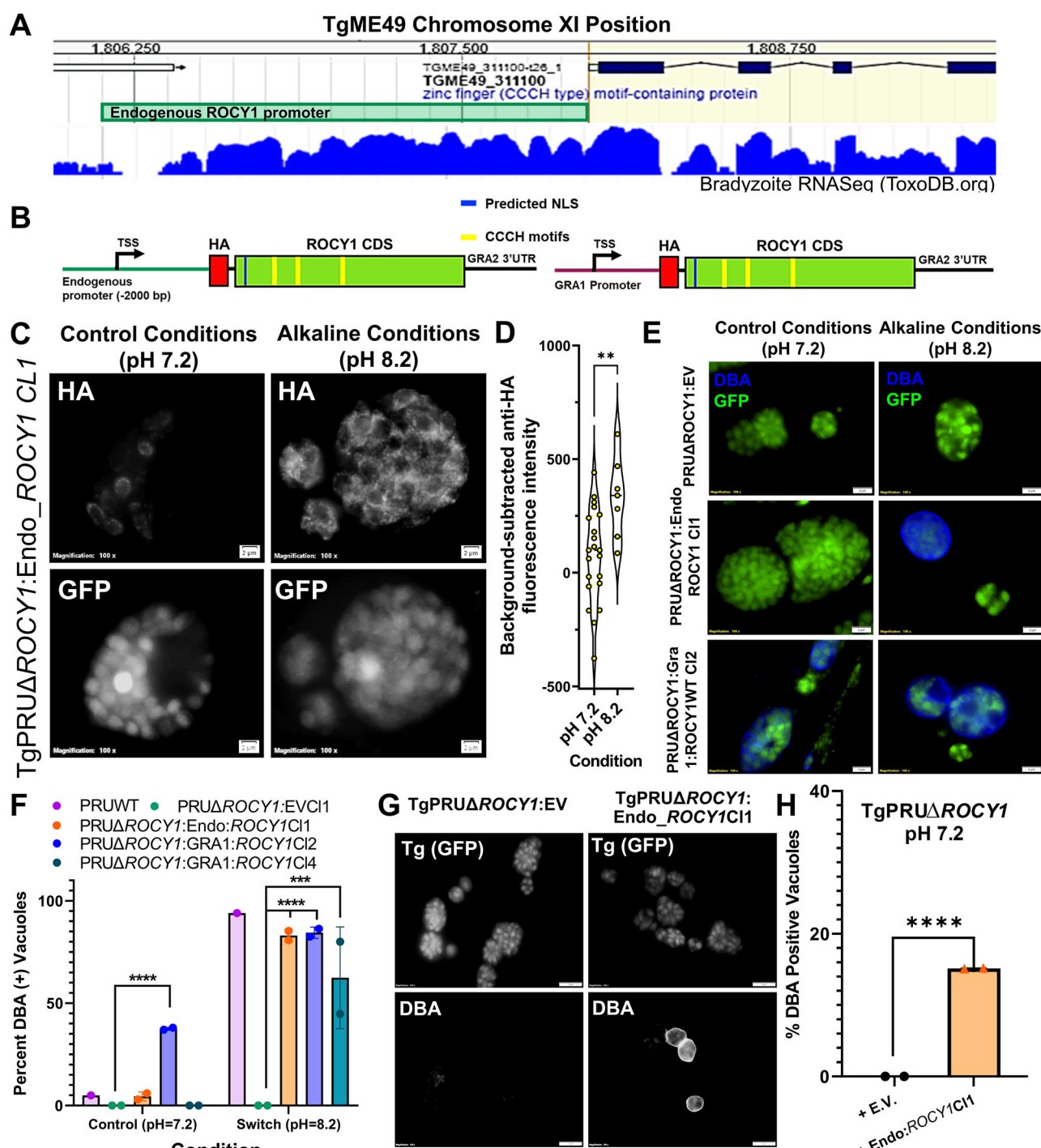

**Fig. 3 | Complementation of Tg PRUΔROCY1 with constructs harboring the endogenous ROCY1 promoter restores the ability to form cysts in response to pH stress in vitro. A**, **B** Schematic of complementation constructs showing the genomic locus for ROCY1 (**A**). **C** Representative images of TgPRUΔ*ROCY1* complemented with ROCY1 promoter-driven constructs in control conditions (pH 7.2) for 3 days or BIC for 4 days. Scale bar: 2 µm. **D** Quantification of change in HA staining intensity in TgPRUΔROCY1 parasites complemented with the N-terminal HA-tagged construct harboring the promoter sequence indicated in **A** and grown under control or high pH conditions. Regions of interest were manually drawn to collect mean HA-staining intensity data and background was determined with a similarly sized region of interest from the same image. **\***: *P* < 0.01 using Student's T-test. **E** Representative images of TgPRUΔ*ROCY1* clones that were transfected with endogenous and overexpression promoter

constructs in control and alkaline conditions. Scale bar: 5 µm. **F** Quantification of the average number of DBA positive vacuoles in 15 parasite-containing fields of view in control and alkaline pH conditions. (*N* = 2 per strain and condition except for wild type, *N* = 1). Statistical significance was determine using a two-way ANOVA with Sidak's multiple comparison test on arcsine transformed data. \*\*\*\**P* < 0.0001. **G** Representative images of spontaneously forming cysts (DBA) in endogenous promoter-complemented TgPRUΔROCY1 but not in parasites complemented with empty vector (EV). Scale bar: 10 µm. **H** Mean cyst formation percentage (*N* = 2 biological replicates per parasite strain) in control conditions in TgPRUΔROCY1:EV and TgPRUΔROCY1 expressing endogenous promoter-driven ROCY1. *P* < 0.0001 based on a Two-tailed Student's T-test on arcsin-transformed data. This experiment was performed once. Data are provided as a Source Data File.

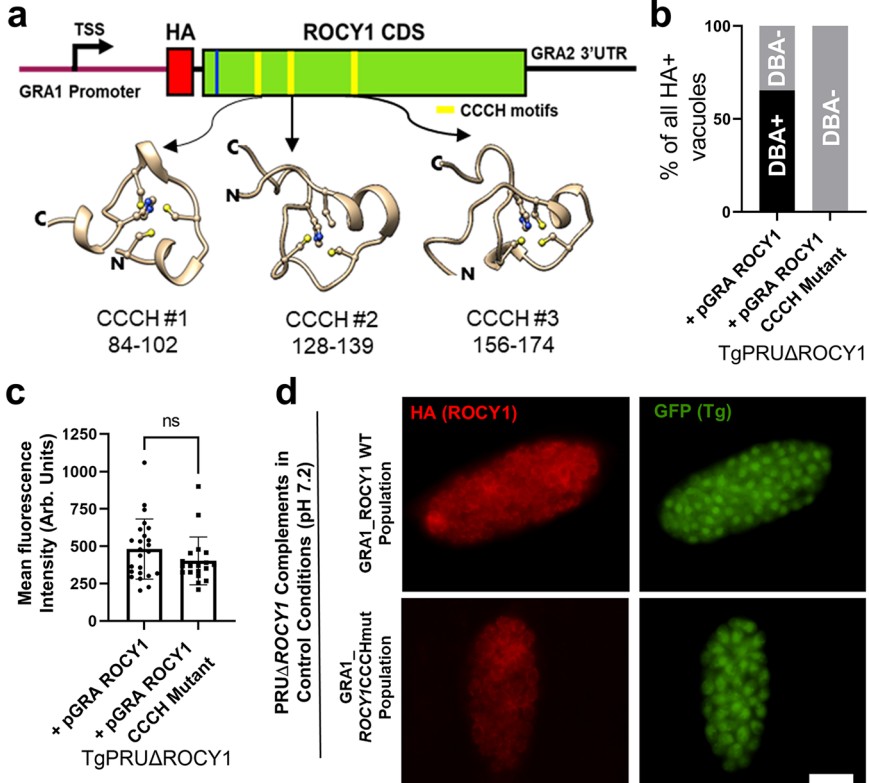

**Fig. 4 | Mutating all 3 CCCH zinc finger binding domains of *ROCY1* abolishes the effect of overexpressing ROCY1 in TgPRUΔROCY1 on cyst formation under control conditions. a** Schematic of the ROCY1 mRNA indicating the location of the putative NLS (blue bar) and CCCH motifs (yellow bars). Shown below are Alpha-Fold2 predictions for the 3 CCCH motifs, with each cysteine and histidine shown with side chains. N- and C-terminal directionality is indicated as are the locations of each CCCH motif. **b** Quantification of the percentage of DBA-positive cysts among HA-positive vacuoles expressing the WT pGRA NHA-ROCY1 ($N = 104$) or the CCCH mutant form of pGRA NHA-ROCY1 ($N = 107$) under control (pH 7.2) conditions. The

CCCH mutations ablated the ability of pGRA NHA-ROCY1 to induce cyst formation in the ROCY1 knockout background. **c** Mean and standard deviation of background-normalized HA-derived fluorescence intensity of TgPRUΔROCY1 expressing either WT pGRA NHA-ROCY1 ($N = 24$) or the CCCH mutant form of pGRA NHA-ROCY1 ($N = 19$) showing similar staining intensity (Student's T-test $P = 0.2$).
**d** Representative images of vacuoles containing PRUΔROCY1 parasites expressing the wild type or CCCHmutant form of the pGra_NHA_ROCY1 plasmid construct. Scale bar: 10 μm (same for all images). This experiment was performed once with this genetic background. Data are provided as a Source Data File.

phenotype after 17 passages in vitro. This is despite consistent expression of GRA1-driven ROCY1 (5 out of 5 clones) and maintenance of their ability to produce cysts in response to BIC (4 out of 5 HA+ clones). We think this may be due to counterselection in vitro for reduced responsiveness to overexpressed ROCY1 and the accompanying slower growth rate although this needs to be further explored. Regardless of the nuances of the phenotypic differences we observed depending on whether the endogenous promoter or GRA1 was used to drive ROCY1 in the knockout background, both clearly complemented the DBA+ cyst forming defect in BIC.

As we examined populations of TgPRUΔROCY1 and passage-matched TgPRU:WT parasites we observed that while wild type parasites formed DBA-positive cysts spontaneously at pH 7.2 at a variable rate between 1 and 10%, we did not observe any DBA-positive cysts under control conditions in TgPRUΔROCY1. To test this hypothesis we counted the number of DBA-positive cysts under normal growth conditions (pH 7.2) for TgPRUΔROCY1:EV and TgPRUΔROCY1:EndoROCY1. TgPRUΔROCY1:EV did not form any DBA-positive cysts across 20 fields of view in each of 2 biological replicates (Fig. 3G, H). In contrast, TgPRUΔROCY1:EndoROCY1 formed cysts spontaneously at a frequency of ~10% (Fig. 3H). This observation, while variable between experiments (for example DBA+ cyst formation under control conditions ranged from 3–6% in the experiment shown in Fig. 3F) suggests that parasites lacking *ROCY1* expression do not spontaneously enter the bradyzoite program (a common feature of many *T. gondii* strains[39,44];).

## At least one of three predicted CCCH-type zinc finger domains are required for ROCY1 induction of DBA+ cyst formation

The *ROCY1* gene is predicted to encode a 920-amino acid protein that contains a predicted nuclear localization signal and 3 CCCH type Zinc finger domains (SMART accession SM000356; Fig. 4a) and AlphaFold2[45] models of each of these motifs suggest that they functional given the orientation of the cysteine and histidine side chains capable of coordinating a single Zinc ion (Fig. 4a). Based on this information we used multiple rounds of site-directed mutagenesis of the pGRA-NHA_ROCY1 construct to mutate 2 of the 3 cysteines to arginine and each histidine to lysine in all 3 CCCH motifs. We transfected TgPRUΔROCY1 parasites with the wild type pGRA:NHA_ROCY1 construct or the pGRA:NHA_ROCY1_CCCH mutant construct and grew them as drug-resistant populations for 2 weeks. For this experiment we examined vacuoles within 3 days of transfection and drug selection rather than cloning since (as described above) we found that parasite lines progressively lost their responsiveness to *ROCY1* overexpression the longer we grew them in culture. When we assessed DBA staining in all HA+ vacuoles in this transfected population we found that 63% of the HA+, TgPRUΔROCY1 vacuoles formed DBA-positive cysts, while not a single parasite vacuole (out of 105) expressing the CCCH mutant form of NHA-ROCY1 was identified as a DBA-positive cyst (Fig. 4b; ChiSquared $P < 0.0001$). Similar results were observed in a second replicate of this experiment (Fig. S3). Based on quantitative immunofluorescence, the difference in cyst formation under control conditions was not due to poor ROCY1 expression as the mean fluorescence intensity was similar between

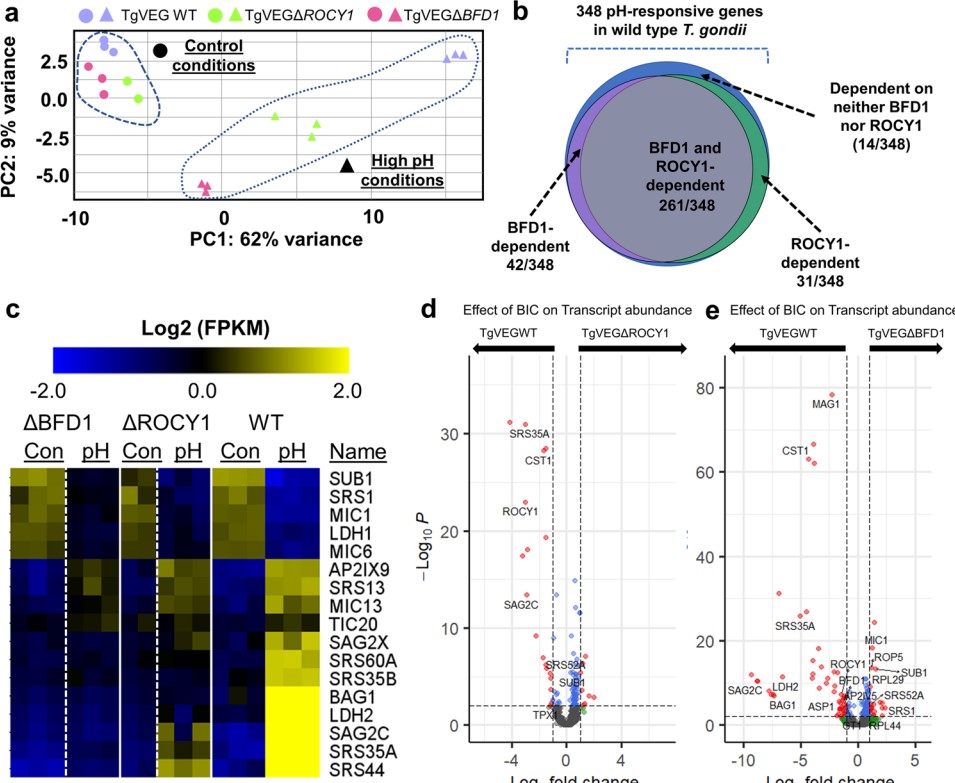

**Fig. 5 | Disruption of the ROCY1 locus dramatically alters the alkaline pH-stress-induced transcriptional response. a** Principal components (PC) 1 and 2 of cells infected with TgVEG WT, TgVEGΔROCY1, and TgVEGΔBFD1 expose to either control (●) or alkaline pH stress (▲) conditions for 48 h. N = 3 per strain and condition. **b** Number of transcripts with significant differences in abundance in high pH conditions compared to control conditions for TgVEG WT, TgVEGΔROCY1, and TgVEGΔBFD1 (|Log₂ Fold Change| ≥ 1, $P_{adj}$ < 0.05). Of the 348 BIC-altered genes, the majority (261) were dependent upon both ROCY1 and BFD1 for their induction (based on a lack of significant induction ($P_{adj}$>0.01) by pH in the knockout parasite line). Data from individual genes along with statistical analyses for pH responsiveness in all three strains is presented in Supplementary Dataset 2. **c** Heatmap of a subset of genes with significantly altered transcript abundance after exposure to high pH in *T. gondii* VEG WT parasites. All genes shown had BIC-altered transcript abundance by |log2FoldChange| > 2 and Padj<0.01. The complete set of BIC-regulated transcripts and their corresponding dependence on the presence of BFD1 or ROCY1 can be found in Supplementary Dataset 2 on the tab titled "348_BIC_IN-DUCED_GENES". **d, e** Volcano plots of transcript abundance differences between wild type TgVEG and either TgVEGΔROCY1 (**d**) or TgVEGΔBFD1 (**e**) under BIC. The majority of transcripts that have significantly different abundance in each knockout line are those induced by BIC in wild type TgVEG. Red datapoints indicate transcripts with significant differences in abundance (Padj<0.01; |log2FC| > 1). This experiment was performed once. Data are provided as a Source Data File.

parasites expressing wild type and CCCH mutant forms of *ROCY1* (Figs. 4c and S3B). Representative examples of HA-positive vacuoles in each population showing the similar staining intensity and localization are shown in Fig. 4d. These data implicate the predicted CCCH motifs as being necessary for ROCY1-driven DBA+ cyst formation in *T. gondii*, and based on the localization of tagged ROCY1 in cytoplasmic puncta as well as the propensity of proteins harboring CCCH-type zinc finger motifs to bind RNA, suggest that interactions with RNA by ROCY1 may be a key to its function in driving cyst formation.

## Disruption of the *ROCY1* locus dramatically alters the alkaline pH/nutrient stress-induced transcriptional response

To better understand the role of ROCY1 in driving pH- and nutrient-stress induced bradyzoite formation, we performed RNAseq on TgVEGΔROCY1, TgVEGΔBFD1, and TgVEG WT parasites exposed to control conditions or BIC for 48 h. Using principal component analysis (PCA) (Fig. 5a), we found that separation along PC1 (62%) represented changes in the parasite transcriptome in response to stress-induced differentiation. All parasites grown in control conditions (WT, TgVEGΔROCY1, TgVEGΔBFD1) clustered together on PC1 and we saw a clear separation between TgVEG WT, and the cyst forming mutant lines (TgVEGΔROCY1 and TgVEGΔBFD1) exposed to BIC on both PC1 (62% of variance) and PC2 (9% of variance; Fig. 5a). TgVEGΔROCY1 parasites fell in the middle between the TgVEG WT and TgVEGΔBFD1 parasites along

PC1, an outcome consistent with the fact that ΔBFD1 parasites have a greater defect in DBA+ cyst formation compared to ΔROCY1 parasites. Raw data for this study can be found in Supplementary Dataset 2.

Based on differential expression analysis with DESeq2 we found that both TgVEGΔROCY1 and TgVEGΔBFD1 parasites grown in control conditions had similar transcriptional profiles to TgVEG WT parasites (all detected transcripts had |Log₂FC| ≤ 1, $P_{adj}$ ≥ 0.01; Supplementary Dataset 2). In contrast the absence of ROCY1 or BFD1 impacted the abundance of hundreds of transcripts after exposure to BIC. For WT parasites, as expected, we identified 348 genes with significant differences in transcript abundance in response to treatment with alkaline stress compared to control conditions (|Log₂FC| > 1, $P_{adj}$ < 0.01) (Fig. 5b and Supplementary Dataset 2). However, the BIC-induced transcriptional response was impaired for both the TgVEGΔROCY1 and TgVEGΔBFD1 parasites, with the majority (261) of the 348 BIC-altered transcripts being dependent on both ROCY1 and BFD1 (Fig. 5b). Well-characterized bradyzoite genes such as CST1 (SRS44; also the binding target of DBA[2];), SAG2C and BAG1 (Fig. 5c−e and Supplementary Dataset 2) were among those that were dysregulated during BIC exposure of ΔROCY1 and ΔBFD1 parasites. Only 14 BIC-altered transcripts were *independent* of both ROCY1 and BFD1 (i.e., they still changed in abundance significantly in the knockout lines when exposed to BIC; Fig. 5b), and some of these transcripts became less abundant after BIC exposure (such as LDH1, SRS1 and MIC1; Fig. 5c). Further evidence for

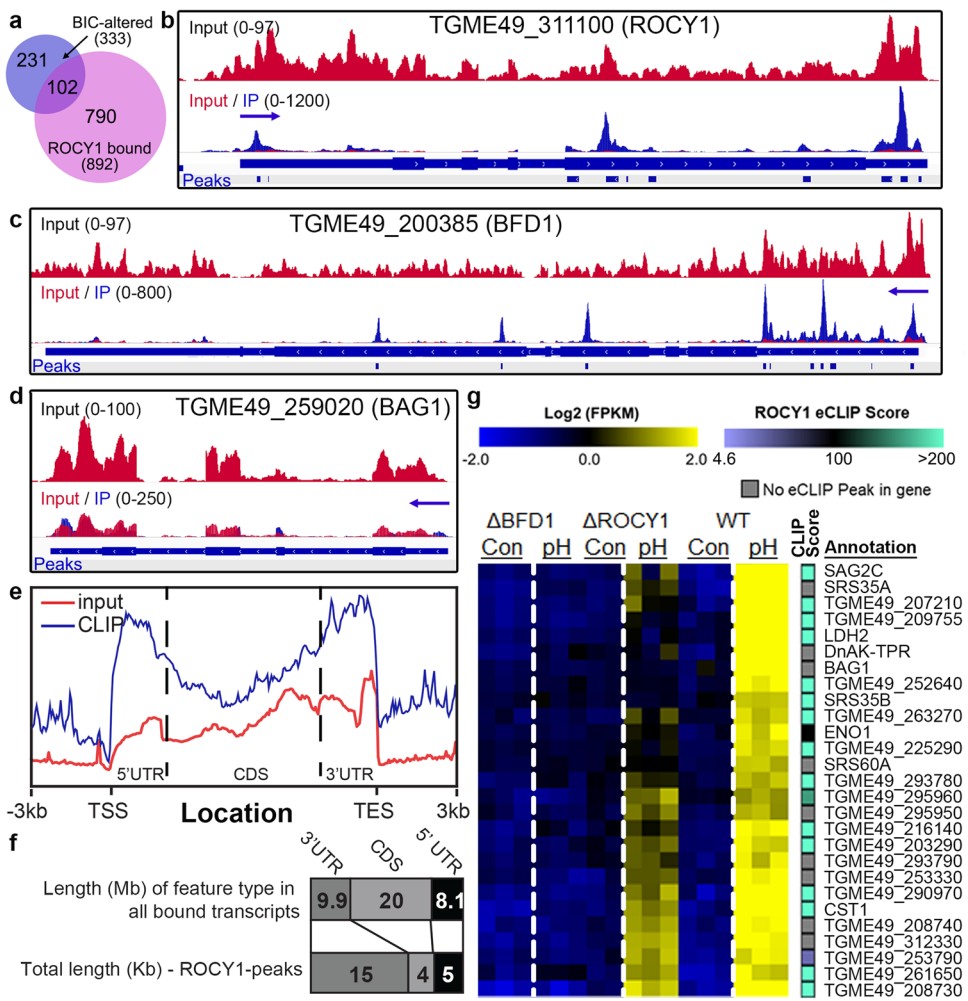

**Fig. 6 | eCLIP analysis of *T. gondii* PRU transcripts bound by N-terminally tagged FLAG-tagged ROCY1.** *T. gondii* PRU parasites were exposed to high pH, cyst inducing conditions for 48 h prior to crosslinking and anti-FLAG immunoprecipitation (*N* = 3). Averages of all 3 libraries are represented here. **a** Venn diagram showing the overlap of: genes having significant changes (|Log2FC > 1|, Padj<0.05) in transcript abundance after exposure to BIC in vitro and found to be significantly bound by ROCY1 based on enrichment in sequenced eCLIP samples. **b** eCLIP data represented on two genome browser tracks showing input (top, red) and input and IP overlaid (bottom, red and blue) and significant ROCY1 peaks associated with the 5' and 3' UTRs and CDS of the ROCY1 transcript. **c** Same as for **b**, but at the BFD1 locus. **d** Same as in **b**, **c**, but for the BAG1 locus. In contrast to the ROCY1 and BFD1 transcripts, no significant ROCY1-bound peaks were detected for the BAG1 transcript. **e** Metagene profile of eCLIP and size-matched input reads spanning from transcriptional start site (TSS) to transcriptional end site (TES). Dotted lines indicate the borders between 5'UTR, CDS and 3'UTR. **f** Quantification of the total length of each transcript feature type (5' UTR, 3' UTR, CDS) in all genes with > 1 ROCY1 peak across the entire transcript (top stacked bar) and across only those regions with significant ROCY1 eCLIP peaks at least 11 bp in size and with a PureCLIP score of at least 50. Chisquare analysis shows a highly significant (*P* < 0.0001), unequal distribution of where the ROCY1 peaks are when normalized to the length of the ROCY1-targeted genes. **g** Mean-centered, clustered heatmap of *T. gondii* genes that are significantly induced by exposure to BIC including both transcript abundance (blue to yellow) and the ROCY1 PureClip score (mauve to teal). Hypothetical protein; MIC: microneme protein. PureClip scores > 4.3 are significant at *P* < 0.01. Grey indicates no eCLIP score above that statistical threshold were detected for that transcript. This experiment was performed once. Data are provided as a Source Data File.

dysregulation of the in vitro pH stress response can be seen in Fig. 5d, e showing differences in transcript abundance between wild type (left arrow) and ether ROCY1 or BFD1 knockout parasites (right arrow) when exposed to BIC for 48 h. Taken together these data indicate that ROCY1 functions as a positive regulator of bradyzoite development and is either found within the BFD1-driven bradyzoite developmental pathway or is a parallel inducer of bradyzoite development involving a very similar collection of downstream transcripts.

## FLAGROCY1 binds multiple mRNAs including its own and has a preference for 3' UTRs

Given the presence of CCCH-type zinc finger domains in ROCY1 and its requirement for bradyzoite formation, we wanted to assess its ability to bind mRNAs in the context of *Toxoplasma* differentiation. We performed enhanced crosslinking immunoprecipitation (eCLIP) from

NFlag-ROCY1 parasites subjected to 48 h of BIC (in biological triplicate; Fig. S4A). eCLIP and its downstream analyses is capable of identifying transcripts that are bound by a protein of interest. We used PureClip[46] to analyze the eCLIP data and found that 6,178 significant FLAGROCY1-crosslink sites were identified in all 3 samples (Fig. S4B, C). These concordant sites were found in a total of 1,061 unique transcripts and had, on average, ~11 crosslink sites per transcript (Fig. S4E). Across all three biological replicates we identified a significant enrichment for crosslink sites in the 3' UTR compared to other transcript regions (Fig. S4F). After using PureClip to merge data from all three biological replicates and individual sites into binding segments we found 3,228 transcript regions across 891 transcripts (Supplementary Dataset 3). Overall, of the 333 transcripts induced by BIC in *T. gondii* (data from Fig. 5), 102 of them had signficant ROCY1 binding sites based on our analysis of eCLIP data (Fig. 6a). The repertoire of ROCY1 RNA targets,

therefore, is relatively large, but there are multiple lines of evidence suggesting that ROCY1 has specific transcript preferences. For example, the *ROCY1* transcript has 2 distinct regions of ROCY1-bound sequence in its 5' UTR, 5 in its CDS and 3 in its 3'UTR (Figs. 6b, S4G). In contrast, *BFD1* has 3 prominent enrichment peaks in its CDS and 7 additional regions of enrichment in its 5' UTR (Fig. 6c). However despite having a ~3 Kb 3' UTR (based on the most up to date community annotation of this gene on ToxoDB.org), we found no enrichment in our pulldowns for this part of the BFD1 transcript sequence (Fig. 6c). Furthermore, ROCY1 does not appear to interact with all highly expressed bradyzoite transcripts. For example, no significant ROCY1-eCLIP peaks were identified in the *BAG1* transcript, despite the fact that it is a canonical marker of late bradyzoite development (Fig. 6d). ROCY1 also has a preference for 5' and 3' UTRs based on metagene plots (Fig. 6e) but if we compare length-adjusted transcript features present in the ROCY1 target pool and the length of the ROCY1 eCLIP enriched regions we see that a preference for 3' UTRs is the most prominent feature of its binding profile (Fig. 6f). Taken together these data strongly suggest that ROCY1 is a functional mRNA binding protein with a slight preference for non-translated regions. We also examined the overlap between genes that are bound by ROCY1 and those that are dysregulated in ΔROCY1 or ΔBFD1 parasites compared to wild type during in vitro stage conversion experiments (shown in Fig. 5). Twenty-seven such genes are shown in Fig. 6g, and 18 of these have highly significant eCLIP scores and 9 of them do not, including canonical bradyzoite markers like *BAG1* and *DnAK-TPR*. Genes dysregulated in ROCY1 mutants with high clipSeq scores included genes like *SAG2C*, *CST1 (SRS44), LDH2* and *SRS35B* (Fig. 6g). These data suggest a degree of specificity in the interactions between *T. gondii* ROCY1 and it's putative target transcripts despite the relatively large number of targets bound by this protein under BIC. We should note that ROCY1 was found to interact with hundreds of putative target mRNAs using RNA immunoprecipitation sequencing (RIP-seq) in a recent study by Licon et al.[33]. As in our study, transcripts like BFD1, CST1 and LDH2 are candidate substrates for ROCY1, while BAG1 was not.

### ROCY1 is necessary for the formation of DBA+ tissue cysts in mice

To determine if ROCY1 was important for DBA+ cyst formation in vivo, we infected CBA/J mice with TgVEG WT-GFP-LUC, TgVEGΔ*ROCY1*-GFP-LUC, and TgVEGΔ*BFD1*-GFP-LUC parasites in two separate experiments and monitored pathogenesis, morbidity and mortality using weight loss and in vivo bioluminescence imaging. In both experiments mice were first imaged at 3 h post-infection and we observed no significant differences in luciferase signal at this time point between strains. In the first experiment parasite burden was indistinguishable during the acute phase of infection (Fig. 7a) across parasite strains, and while parasite mortality was statistically indistinguishable between strains (Fig. S5A), we did observe minor, but significant, differences in weight loss in mice infected with ΔROCY1 or ΔBFD1 parasites (Fig. 7b). We sacrificed mice at 4 weeks post-infection and used a portion of the brain to count GFP+/DBA+ cysts and reserved the rest for other manipulations (below). Mice infected with WT parasites harbored an average of 168+/−80 DBA+ cysts/brain (Fig. 7c), while mice infected with TgVEGΔROCY1 had significantly fewer detectable DBA+ cysts (2.8+/−5; $P = 0.0014$), and DBA+ cysts were not detected at all in 6 of 8 surviving mice. The cysts that we did identify stained positive for DBA lectin in a fashion similar to wild type, indicating that ΔROCY1 parasites are still capable of forming a small number of DBA+ brain cysts (Fig. 7d). TgVEGΔBFD1 parasites were also deficient at forming DBA+ cysts in vivo (as expected[10]) and we were unable to detect any TgVEGΔBFD1 DBA+ cysts in any infected mice ($N = 8$; Fig. 7c). Despite the large difference in DBA+ cyst burden between mice infected with ROCY1 or BFD1 knockouts compared to wild type, mice infected with

all 3 parasite lines had similar numbers of parasite genome equivalents in the brain at this time point (Fig. 7e). For the second experiment we again found that parasite burden was similar between strains during the acute phase of infection, although we did detect significantly lower burden in TgVEGΔROCY1 parasites compared to wild type controls on days 7 and 8 post-infection and some differences in mouse weight on days 16 and 18 post-infection in TgVEGΔBFD1-infected mice (Fig. S5B, C). We harvested brains and counted DBA+ cysts at either 3 or 9 weeks post-infection and found no identifiable DBA+ cysts in mice infected with both knockout strains (in contrast to wild type; Fig. S5D, F). In brains from 3 weeks post-infection we found similar numbers of parasite genomes in mice infected with all 3 parasite lines (Fig. S5E), while at 9 weeks post-infection mice infected with ΔROCY1 and ΔBFD1 had detectable *T. gondii* DNA equivalent to hundreds of parasites, but mice infected with wild type mice had significantly more (by at least an order of magnitude) *T. gondii* genome equivalents (Fig. S5G). Overall these data suggest that ΔROCY1 and ΔBFD1 are capable of persisting in infected mice for up to 9 weeks post-infection, but that they decline in number in the later stages of chronic infection.

In order to ensure that we could restore the in vivo DBA+ cyst formation capacity in *T. gondii* parasites lacking *ROCY1*, we infected mice with wild type TgPRU:WT ($N = 4$), TgPRUΔ*ROCY1* ($N = 4$) and TgPRUΔ*ROCY1*:Endo*ROCY1* ($N = 5$) strains described above (Fig. 3) and quantified cyst burden in the mice after monitoring parasite burden using in vivo bioluminescence, viability and weight loss (Fig. S6A–C). Mice infected with wild type and complemented parasites both had significantly greater numbers of cysts compared to the PRUΔROCY1 parental line (Fig. S6D), and we detected a small number of DBA+ cysts in the brains of 3 of the 4 mice infected with TgPRUΔROCY1 (Fig. S6D). Of the cyst-like structures identified in mice infected with TgPRUΔR-OCY1 they all stained only weakly with DBA (Fig. S6E), although the number of cysts observed in this particular experiment did not permit quantification of this phenotype. Example images of in vivo cysts from each parasite line are shown in Fig. S6E.

### In vivo, *T. gondii* cyst forming mutants have biological features in vivo that are not typically associated with bradyzoites and/or cysts

To directly observe persisting forms of *T. gondii* mutants we stained brains from mice infected for 4 weeks and were able to easily find foci containing *T. gondii* parasites across all brains regardless of whether the mouse was infected with TgVEG WT, TgVEGΔROCY1 or ΔBFD1 strains (Fig. 7f). Across 2 random sections from the brain of each mouse we detected foci of parasites in 7/9 mice infected with WT parasites, 5/8 mice infected with TgVEGΔROCY1, and 5/8 mice infected with TgVEGΔBFD (Fig. 7f). While the vacuole phenotype was variable across sections and individual mice, we did observe a trend that the vacuoles in ΔROCY1 and ΔBFD1-infected mice were generally smaller and often found in localized foci of multiple vacuoles moreso than in mice infected with WT parasites (Fig. 7f). This suggested to us that TgVEGΔROCY1 and TgVEGΔBFD1 parasites were perhaps persisting in the brain at 4 weeks post-infection in a state more reminiscent of tachyzoites than bradyzoites, although it is still possible that they are surrounded by a cyst wall of some kind that just binds the DBA lectin very poorly. In albino Balb/C mice in vivo bioluminescence imaging can be used to quantify parasite burden in the brain, and in multiple studies we and others[47–49] have found that signal arising from the head of the mouse is readily detected during the transition between the acute and chronic phases of infection (typically during days 10–14[49,50]). We infected mice with either TgVEG:WT or TgVEGΔROCY1 parasites and imaged them dorsally over the course of the infection. We observed a consistently higher luminescent signal in the brain of TgVEGΔROCY1-infected mice starting on day 13 post-infection (quantified in Fig. 7g), and found that detectable luminescence persisted in TgVEGΔROCY1-infected mice for at least 33 days (Fig. 7g). This was in

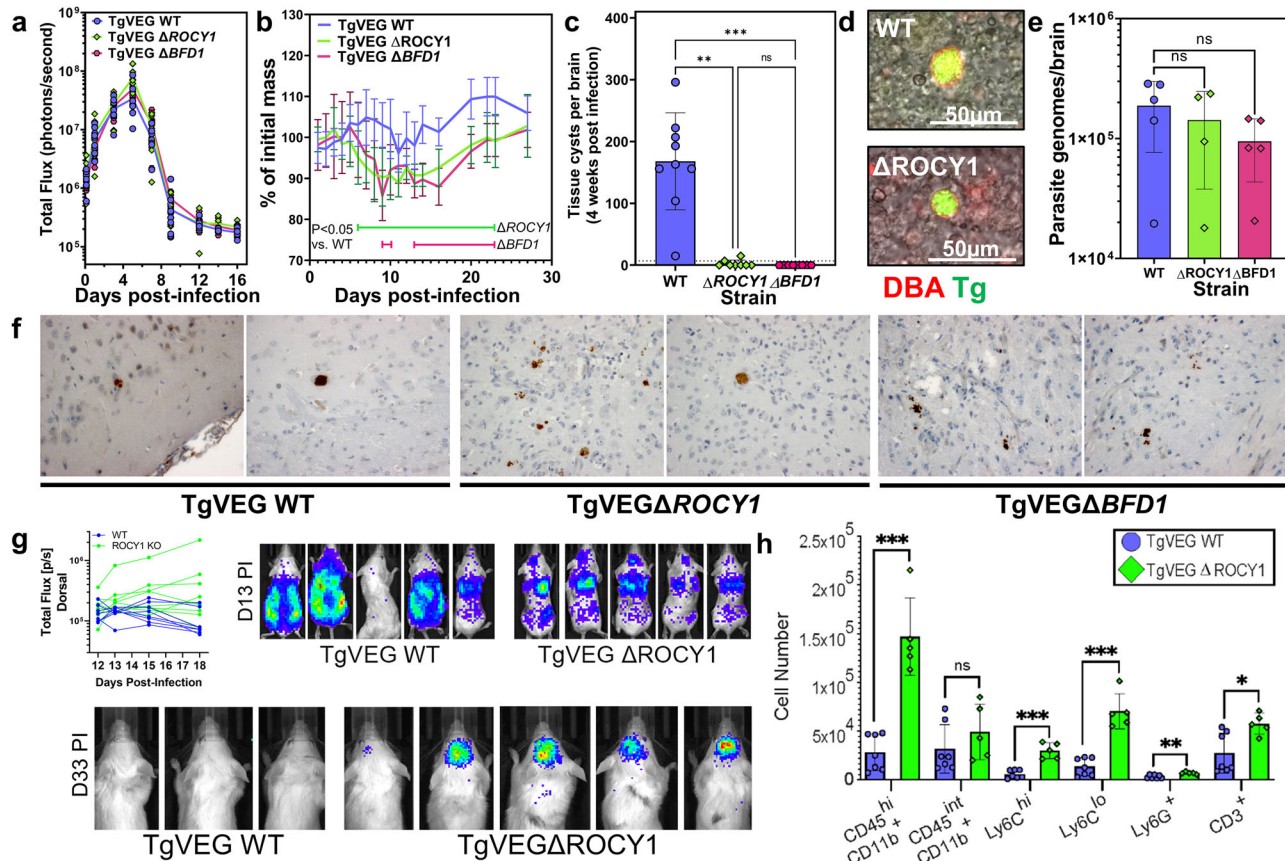

**Fig. 7 | ROCY1 is necessary for tissue cyst formation in mice but not for parasite persistence. a** Quantification of bioluminescent imaging of mice infected with TgVEG WT-GFP-LUC, TgVEGΔ*ROCY1*-GFP-LUC, and TgVEGΔ*BFD1*-GFP-LUC parasite (*N* = 6). Each point represents an individual mouse and the line represents the mean. Statistical significance determined by two way ANOVA and Tukey's Multiple Comparison post-test. **b** Mean and standard deviation of weight loss over the course of the experiment in A, with statistical significance assessed by two Way ANOVA and Tukey's Multiple Comparison post-test; *N* = 10 mice per parasite strain. **c** Brain tissue cyst burden observed at 4 weeks post-infection for experiment in A. Mean and standard deviation are plotted and each point represents and individual mouse. Statistical significance was determined using a Kruskal-Wallis test with Dunn's multiple comparisons (***: *P* ≤ 0.001; **: *P* ≤ 0.01; (limit of detection = 7 cysts). **d** Representative images of TgVEG:WT and TgVEGΔROCY1 brain cysts stained with rhodamine-labeled DBA. **e** Quantification of parasite genomes per brain in infected mice at 4 weeks post infection. Mean and standard deviation are

plotted. Each point represents an individual mouse. Statistical significance was determined using a one-way ANOVA with Tukey's multiple comparison test on ΔC$_t$ values. **f** Immunohistochemistry from intact brains using anti-GFP antibodies to localize *T. gondii* parasites taken from mice infected with the indicated strain for 30 days. Two representative images from a single mouse for each infecting strain are shown. **g** Quantification of of in vivo bioluminescence imaging of mice infected with TgVEG WT or TgVEGΔROCY1 parasites and representative images showing higher dorsal bioluminescent signal in mice infected with ROCY1-knockout parasites. Data were quantified on days 12–18 and images are shown for days 13 and 33 postinfection. *N* = 8 mice for WT and *N* = 9 mice for ΔROCY1. **h** Flow cytometric quantification of immune cells (mean +/- SD) in the brain of WT and ROCY1 knockout parasites at day 21 post-infection in the brains of mice infected with TgVEGΔROCY1 (*N* = 7) compared to TgVEG:WT (*N* = 5). This experiment was performed twice and data from both experiments are pooled. Data are provided as a Source Data File.

stark contrast to wild type *T. gondii* VEG, which we failed to detect in the head area at 33 days PI (Fig. 7g). Taken together these data illustrate a distinct outcome for the transition to the chronic phase of infection in the ΔROCY1 and ΔBFD1 parasite lines and this suggests an in vivo failure to successfully transition into the slower growing, bradyzoite form.

Since tachyzoites have a characteristic inflammatory signature that is distinct from slowly bradyzoites, we subjected subsamples of brains from 4-week infected mice (*N* = 5 per parasite strain) to RNA sequencing and compared both host and parasite gene expression profiles using DESeq. Using PCA, TgVEGΔROCY1 and TgVEGΔBFD1 brain RNAseq samples separated from wild type samples on the main principal component (Fig. S7A) and were similarly adjacent based on unsupervised clustering (Fig. S7B). We identified 1189 transcripts that were of significantly different abundance in mouse brains infected chronically with TgVEGΔROCY1 compared to those infected with TgVEG:WT and most (851) of these were shared between mice infected with both knockout parasite lines (Fig. S7C, D; Supplementary

Dataset 4). The inflammatory signature of mice infected with ΔROCY1 and ΔBFD1 parasites was also significantly different as determined using gene set enrichment analysis (GSEA[51];), where pathways like "HALLMARK INFLAMMATORY RESPONSE" and "HALLMARK IL2 STAT5" were similarly and significantly enriched based on a false-discovery rate of 0.05 (Fig. S7D, F and G). This difference in immune response find support in a parallel analysis using Cibersort[52] which infers percentages of different cell populations in bulk RNAseq data and has been applied to brain RNAseq data from *T. gondii*-infected mice[53]. Given our interest in the inflammatory profile of mice infected with wild type and cyst-deficient *T. gondii* (ΔROCY1 and ΔBFD1), we used the precomputed "LM22" as the signature gene file and used it to estimate the relative proportions of 22 immune cell types using our brain RNAseq data. Of the 22 cell types queried CD8 + T cells were predicted to be significantly higher in abundance in both ΔROCY-infected (*P* < 0.0001) and ΔBFD1-infected (*P* < 0.01) mice based on brain RNAseq data (Fig. S7E). These data are consistent with the idea that by 4 weeks post-infection wild type TgVEG parasites are causing

significantly less inflammation compared to ΔROCY1 and ΔBFD1 parasites, and that there may be more CD8 + T cells in the brains of mice infected with the mutant parasites to combat was is being sensed as an acute, rather than chronic, infection.

To examine the state of the immune response in the brain at the cellular (rather than transcriptomic) level, we infected C57BL/6 mice with TgVEG WT or TgVEGΔROCY1 parasites and quantified immune cell populations in the brain using flow cytometry at day 28 post-infection. We found significantly increased levels of immune cell infiltration − including greater numbers of myeloid cells (CD45$^{hi}$CD11b$^+$), Ly6C$^{hi}$ monocytes (CD45$^{hi}$CD11b$^+$Ly6C$^{hi}$), Ly6C$^{lo}$ monocytes (CD45$^{hi}$CD11b$^+$Ly6C$^{lo}$), neutrophils (Ly6G$^+$), and T cells (CD3$^+$) − but no changes in the brain resident microglial levels (CD45$^{int}$CD11b$^+$) in VEGΔROCY1-infected mice compared to TgVEG WT-infected mice (Figs. 7H, S7H).

Finally, we subjected a subset of brain RNA samples from mice infected for 30 days with wild type or ΔROCY1 parasites ($N$ = 3 per parasite strain) to deep, dual-RNAseq to identify differences in the parasite transcriptonal profiles during the chronic phase of infection. After filtering for low read counts we were able to quantify abundance for 1,225 *T. gondii* transcripts (Supplementary Dataset 4). Multiple canonical bradyzoite markers were of significantly lower abundance in brain-resident ΔROCY1 parasites, including BAG1, LDH2, BRP1 and AP2IX-9 (Fig. S8A, B). Moreover, a number of genes more typically associated with tachyzoite replication including multiple dense granule (GRA1, 2, 3, 5 and 6) and rhoptry (ROP1, RON8) protein-encoding genes are of significantly higher abundance in brain-resident ΔROCY1 parasites (Fig. S8A, B). Not all genes behaved similar in vivo compared to in vitro. For example MCP4 and MAG1 were both of significantly lower abundance in *ROCY1* and *BFD1* knockouts after exposure to BIC (Fig. 5 and Supplementary Dataset 3) but were found to be of significantly greater abundance in the *ROCY1* knockouts in vivo (Fig. S8B; Supplementary Dataset 4). When we examined the densities of Log2FC values based on whether the genes had been categorized as being "tachyzoite" or "bradyzoite"-specific[29] we saw a clear and significant (two-sided Kolmogorov-Smirnov test $P$ = 1.6 × 10$^{-8}$) difference in these densities consistent with brain-resident, day 30 ΔROCY1 parasites having a more tachyzoite-like transcriptional profile (Fig. S8C).

### *Toxoplasma* parasite mutants with defects in DBA+ cyst formation in vivo are as capable of reactivation as wild type parasites for up to 5 months post-infection

Bradyzoites have been logically invoked as the culprit for causing recrudescent disease in animal models and in immune compromised human patients, whereby cysts somehow rupture either randomly or due to a lack of immune surveillance and then are able to cause lethal, disseminated disease, as the host is now poorly protected against the infection. Under this model, one would expect slower or reduced recrudescence by parasites that have defects in tachyzoite to bradyzoite conversion and, at a minimum, altered cyst wall protein composition. To test this hypothesis we infected mice with TgVEG WT, TgVEGΔ*ROCY1*, and TgVEGΔ*BFD1* parasites and gave them dexamethasone in their drinking water starting on day 30 post-infection. Recrudescence was monitored using bioluminescence imaging[47,54] on multiple days after dexamethasone treatment (Fig. S9A, B). We performed this experiment twice with similar sample sizes and time-points. For experiment 1 (Exp1) we observed a significant increase in luciferase signal in dexamethasone-treated mice infected with TgVEG WT parasites as compared to non-treated control mice on all time-points starting at D4 post-dexamethasone treatment. We observed evidence for reactivation with similar kinetics in mice infected with TgVEGΔ*ROCY1* (days 4, 7, 8 and 10) and with TgVEGΔ*BFD1* (all days between 4 and 14; Fig. S9A, B). All mice in this experiment that were treated with dexamethasone became morbid (Fig. S9D), while all

infected mice given normal drinking water survived (Fig. S9D), but we did find that dexamethasone treated mice infected with TgVEGΔ*BFD1* died significantly sooner than those infected with TgVEG WT ($P$ = 0.002; Fig. S9D). We repeated this experiment a second time and obtained similar results. Again *T. gondii* parasites reactivated with similar timing and magnitude regardless of parasite strain (Fig. S9C), showing significantly higher luminescence on D11 for TgVEGΔ*ROCY1* and on days 7 onward for TgVEGΔ*BFD1* and TgVEG:WT. Mice infected with both TgVEGΔ*ROCY1* or TgVEGΔ*BFD1* succumbed to the infection significantly sooner ($P$ = 0.004 for both strains) compared to those infected with wild type parasites (Fig. S9F). For both experiments overall weight loss generally tracked with morbidity (Fig. S9E, G) although we did not quantify this statistically.

In a third in vivo experiment we tested the reactivation kinetics in mice 5 months post-infection. Again, we found that all 3 strains were capable of reactivating at this late time point within 8 days after the start of dexamethasone treatment (Fig. 8a, b) as demonstrated by having significantly greater luminescence signal at post-dexamethasone time points (Fig. 8b). Wild type *T. gondii* VEG did not exhibit any advantage in its ability to reactivate compared to ΔROCY1 or ΔBFD1 across all 3 separate experiments, showing that *T. gondii* parasites with clear defects in bradyzoite formation and, at a minimum, lower levels of DBA-binding proteins in the cyst wall, in vitro and in vivo are just as capable of causing recrudescent disease as their wild type counterparts.

Finally we aimed to determine the infectivity of brain-resident wild type and mutant parasites isolated during the chronic phase of infection. Since the cyst wall is thought to be critical for oral infectivity of *T. gondii*, if the cyst wall is impaired in ROCY1 and BFD1 mutants this should lead to reduced oral infectivity compared to wild type parasites. We harvested the brains and liver of mice infected for 198 days ($N$ = 4 per strain; (Fig. 8c)) and used them in multiple assays. All three brain homogenates contained detectable parasite DNA (Fig. 8d) although significantly less DNA was detected in the brains of ΔROCY1 and ΔBFD1-infected mice (similar to the experiment performed above at 9 weeks post-infection; see Fig. S5). All mouse brains contained live parasites, as they could be grown in vitro directly from the brain homogenates either immediately (for wild type) or after a single passage (for ΔROCY1 and ΔBFD1; Fig. 8e). Live parasites could also be isolated from the livers of 2/4 mice infected with wild type parasites, while no parasites grew from any of the livers harvested from mice infected with ΔROCY1 or ΔBFD1 after as many as 5 passages (Fig. 8e). When we gavaged naïve recipient mice with 1/6$^{th}$ of the brains from the infected mice, TgVEG:WT brain homogenates induced detectable serum IFNγ by D6 post-infection while we detected no IFNγ in mice orally infected with mutant parasite brain homogenates (Fig. 8f). Serum taken from D49 post-gavage in these same mice showed efficient seroconversion in wild-type infected mice while mutant infected mice failed to produce any detectable anti-*Toxoplasma* IgG. (Fig. 8g). Finally, we readily cultured live *T. gondii* from the brain and lungs (but not the liver) of the one wild-type infected mouse that survived the infection, but failed to do so in any of the mice gavaged with mutant-infected brain homogenates. While the input number of parasites were lower for mice gavaged with mutant parasites (Fig. 8d), the complete lack of seroconversion suggests reduced oral infectivity since there were hundreds of parasite genome equivalents in the gavaged brain homogenates (Fig. 8d), all homogenates contained live parasites, and even 10 orally administered bradyzoites are sufficient to cause seroconversion (for example, ref. 55). Taken together, these data provide further support that that the cyst wall in BFD1 and ROCY1 mutant parasites has defects severe enough to reduce its oral infectivity. Whether this is due to the complete lack of a cyst wall or rather the presence of a fragile one remains to be determined.

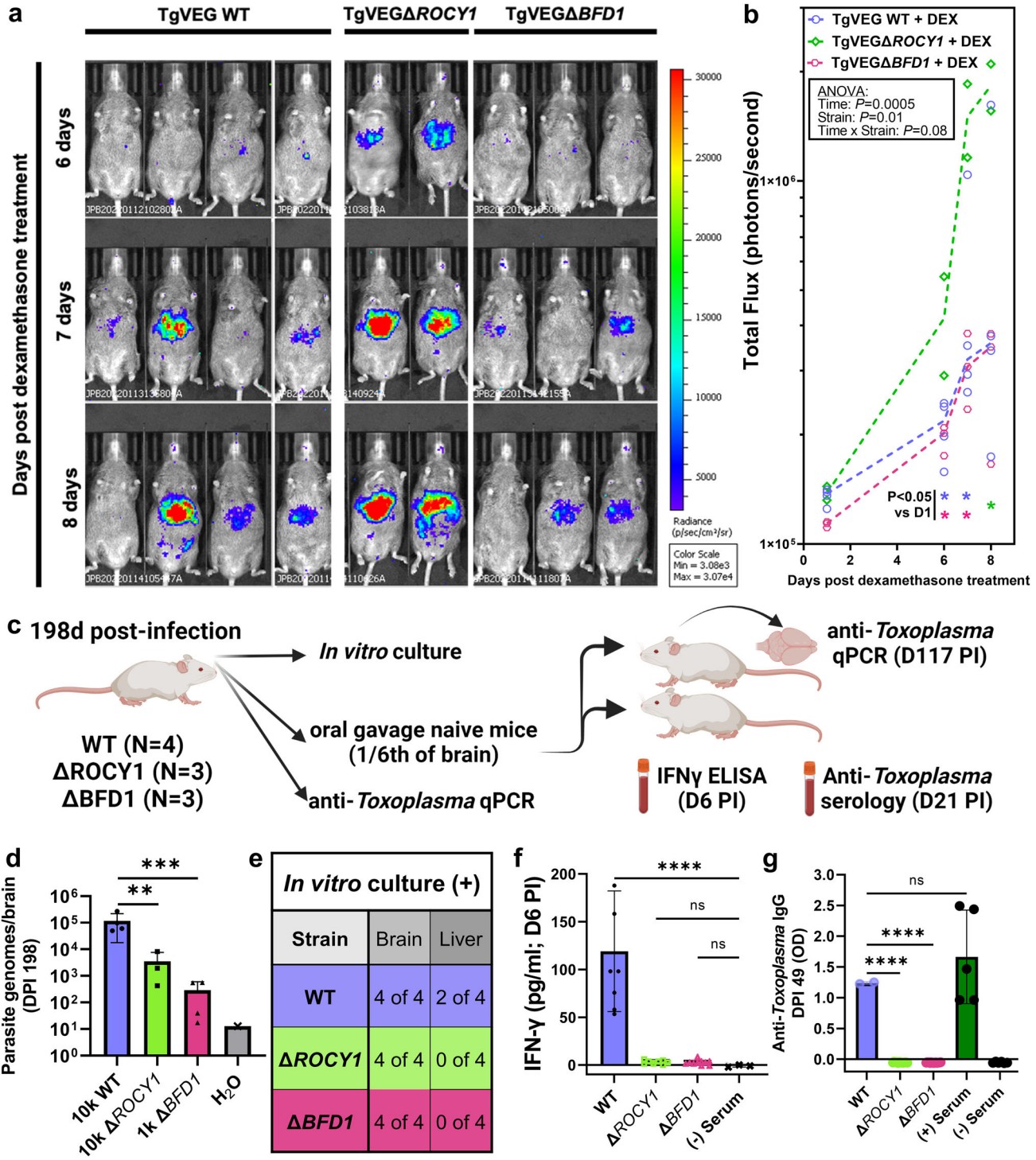

**Fig. 8 | TgVEGΔROCY1 and TgVEGΔBFD1 parasites can be reactivated following chronic infection. a** Select bioluminescent images from the 5 month reactivation experiment showing parasite-derived luciferase signal at days 6, 7, and 8 post-dexamethasone treatment (days 153, 154, and 155 post infection, respectively) in mice infected with TgVEG WT (*N* = 4), TgVEGΔROCY1 (*N* = 2), and TgVEGΔBFD1 (*N* = 3). **b** Quantification of bioluminescent imaging data in (A). Asterisks indicate significant (*P* < 0.05) differences in luminescent signal within each parasite strain when compared to the D1 postdexamethasone treatment timepoint using a 2 way ANOVA and Fisher's Exact LSD test. Only pre-planned comparisons were performed to minimize Type I Error. Lines shown connect the median values for each time point and ANOVA *P*-values are also shown. **c** Schematic of experiment designed to

determine persistence and oral infectivity of wild type and mutant parasites on day 198 postinfection. Created with Biorender.com. Brains of surviving mice were inoculated orally into 2 recipient mice and infectivity was tested by in vitro cultivation (**e**), serum levels of IFNγ (**f**; D6 PI; *N* = 6 mice per group except negative controls which had *N* = 3) and serum anti-*Toxoplasma* antibodies (**g**; D49 PI; *N* = 2 for WT, 12 for ΔROCY1, 16 for ΔBFD1, 5 for + control and 6 for − control serum). Data in **d**, **f** and **g** are shown as mean +/- standard deviation and analyzed by one way ANOVA followed by a Holm-Sidak Multiple comparison post-test comparing indicated groups to wild type. ***: *P* < 0.001; ****: *P* < 0.0001. Sample sizes for **d** are shown in panel **c**. This experiment was performed once. Data are provided as a Source Data File.

## Discussion

The conversion from fast-growing tachyzoites to slow-growing bradyzoites is associated with establishment of chronic *T. gondii* infection and has long been thought to be fundamentally important for long-term persistence and ultimately transmission to the next host[17,56,57]. Reactivation of dormant stages upon immunosuppression is a sensitive measure of parasite persistence, where even very small numbers of persisting parasites can be detected after their reactivation using in vivo imaging or progression of infected animals towards morbidity. We reasoned that recrudescence studies with mutant parasites that are severely impaired in their ability to form DBA+ cysts in vitro and in vivo would be an ideal way to probe the link between tachyzoite to bradyzoite conversion and the persistence of parasites that are capable of reactivation. Our work shows that ΔROCY1 and ΔBFD1 parasites do not persist as well as wild type given the significantly reduced brain parasite burden at 9 weeks and ~6 months post-infection. We were therefore surprised that parasites lacking ROCY1 and BFD1 were as capable of reactivating and causing severe disease as wild type parasites. To our knowledge, no other reactivation studies have been performed with parasites that are as severely impaired in their cyst forming capacity as ΔROCY1 and ΔBFD1 parasites, and our observations call into question the importance of bradyzoite development itself in the ability of parasites to persist and ultimately reactivate when immune surveillance is dysregulated by drugs or disease. Our data do not conclusively show that TgVEGΔROCY1 and ΔBFD1 completely lack a cyst wall when they are left undisturbed in situ, but at a minimum most (for ΔROCY1) or all (for ΔBFD1) of the brain-resident parasite-containing vacuoles are disrupted by the cell-straining process used to quantify brain cysts in situ (a standard approach for *T. gondii*[50];). It should be noted that we did identify two types of cyst-like structures in TgPRUΔROCY1 brain homogenates: one that stained well with DBA and another type that did not (but which still appeared to be surrounded by a cyst wall-like structure; see Figs. S6 and 7D). The frequency of each of these types was still much lower than "normal" cysts from wild type parasites, but they suggest that, in contrast to ΔBFD1 parasites, ΔROCY1 parasites still have some capacity to form cyst, or cyst-like, structures, albeit at a much lower frequency than wild type. Importantly these parasites were poorly infectious to naïve mice by oral gavage, despite growing readily in vitro after a single passage.

If formation of a cyst wall that survives the mild processing we use to isolate cysts from mouse brains was important in the ability of *T. gondii* parasites to cause recrudescent disease, we should have at least observed that reactivation occurred more quickly and/or with more pathology in wild type parasites compared to our mutants. As shown in multiple experiments, this is not the case. TgVEGΔ*ROCY1* and Δ*BFD1* parasites did not have ANY deficiencies in their ability to cause recrudescent disease compared to wild type parasites but were, in fact, capable of reactivating with equal speed and ferocity in a manner that was independent of DBA+ brain cyst numbers or parasite genome equivalents by qPCR. The simplest conclusion to make from these data is that the formation of a cyst wall in vivo that survives the processing of the brain tissue required to observe and count it is not required for parasite recrudescence. It could also be that the brain resident parasites (whether contained by a cyst wall or not) are not responsible for recrudescence, and instead life stages found in other tissues (liver, spleen, muscle, and lymph nodes) are responsible. This is consistent with our observation of reactivation occurring in and around the mouse liver (Figs. 8a and S9A), but is inconsistent with failed detection of any live Δ*ROCY1* or Δ*BFD1* parasites in the liver during late stage (i.e., 6 months) chronic infection (as shown in Fig. 8e). The location of recrudescing mutant parasites remains unknown but our data suggest that they may be much more rare than previously thought.

Our host RNAseq and flow cytometric data identified increased abundance of inflammatory cells and inflammatory transcriptional signatures in mice chronically infected with mutant parasites compared to WT. Our parasite transcripomics showed that in addition to being poor formers of DBA+ cyst walls *T. gondii* Δ*ROCY1* brain parasites have a transcriptional profile much more similar to tachyzoites than bradyzoites. It is possible, therefore, that cyst formation and/or even the transition from the tachyzoite to the bradyzoite is not absolutely required for *T. gondii* parasites that are ultimately capable of recrudescence to survive for long periods of time in the infected host. Our data suggest that recrudescing parasites may represent a distinct subpopulation of parasites, a state that would be analogous to the hypnozoite in *Plasmodium vivax* and other forms of malaria that cause recrudescent disease[58]. If proven to be the case in future studies examining the mutant cyst wall more thoroughly (for example via electron microscopy or development of gentler purification methods), this would represent a significant change in our understanding of *T. gondii* persistence and the *T. gondii* life cycle. Most (but not all) studies focused on *T. gondii* persistence over time and/or after exposure to putative cyst-targeting drugs restrict their analysis to parasites bound by a robust cyst-wall.

Our in vitro cyst induction assays and transcriptomics analyses demonstrate that ROCY1 is critical for DBA+ cyst formation and for the induction of multiple well-known bradyzoite-associated transcripts, such as BAG1, LDH2, SAG2C, and SRS35B (Fig. 5e). We identified extensive overlap in the dysregulated genes in TgVEGΔ*ROCY1* parasites as compared to TgVEGΔ*BFD1* parasites in vitro, suggesting a link between the function of these two genes, a premise further supported by the presence of multiple BFD1 binding motifs in the promoter of ROCY1, which we was shown in published BFD1 Cut&Run data[10]. However ROCY1 does not appear to operate strictly downstream of BFD1, given that ROCY1 overexpression is sufficient to induce cyst formation under control conditions. ROCY1 may stabilize and/or promote the translation of the BFD1 transcript, a mechanism that finds support in our eCLIP, overexpression and site-directed mutagenesis studies. Moreover, a study recently published on ROCY1 (named BFD2[33]) deletion of ROCY1 leads to decreased levels of nuclear BFD1, a finding that fits the model that ROCY1 binds to the *BFD1* transcript to promote its stability and/or translation. Consistent with our ROCY1 eCLIP data, it was also shown in that same study that ROCY1 specifically bound hundreds of transcripts including that for BFD1[33]. Other *T. gondii* RNA binding proteins, such as TgAlba1 and TgAlba2, are involved in bradyzoite development and function in translational control[42], although it should be noted that the TgΔAlba1 cyst formation phenotype in mice was not as dramatic as that observed here for TgVEGΔ*ROCY1*. In other systems, CCCH zinc fingers have been found to promote RNA deadenylation and degradation by binding to AU rich elements in the 3' UTRs of mRNA transcripts[59]. Given that ROCY1 binds to different regions depending on the transcript (for example in the CDS and 5' UTR of BFD1 and primarily the 3'UTR of ROCY1) the mechanism of action could differ depending on the location of binding, with 3'UTR binding promoting mRNA stability and CDS/5'UTR binding promoting translation. The impact of ROCY1-mRNA interactions may be quite vast, given that ROCY1 eCLIP pulldown samples were enriched for nearly 1000 *T. gondii* mRNAs. While this number may appear to be quite large (this represents ~1/10th of the *T. gondii* transcriptome), RNA binding proteins in other systems have been shown to have similar broad mRNA interaction profiles. These include ELAVL1[60], the CCCH zinc finger domain containing PRRC2B[61] and MEX3A[62], all of which have hundreds to thousands of putative mRNA targets.

Taken together, these findings demonstrate that ROCY1 is an important parasite factor for bradyzoite development in *T. gondii* and contributes to the activation of a gene regulatory network used by these parasites to initiate the changes required for effective stage conversion. *ROCY1* mutants largely phenocopy *BFD1* mutants in both their stage conversion capacity in vitro and in vivo and ability to reactivate upon immunosuppression. These data raise important

questions about the importance of stage conversion as the field defines it in the ability of parasites capable of recrudescing to persist for many months in the infected host.

## Methods

### Host cell and parasite strains

*Toxoplasma gondii* strain VEG (TgVEG)[63] and *Hammondi hammondi* strain HhCatAmer (HhAmer)[15] were isolated from oocysts from experimentally infected cats provided by J.P Dubey as previously described[16,64]. The *Toxoplasma gondii* strain CEPΔHXGRPT-GFP-LUC (TgCEP)[65] was also used. All *T. gondii* parasites were maintained through passage in Human foreskin fibroblasts (HFFs) cultivated with DMEM supplemented with 100 U/mL penicillin, 100 μg/mL streptomycin, 2mM L-glutamine, and 10% FBS (cDMEM) and maintained at 37 degrees C, 5% CO2.

### Oocyst excystation

Sporozoites were harvested from oocysts as previously described[16,64]. Following excystation, sporozoites were used to infected confluent monolayers of HFFs grown in T-25 tissue culture flasks and incubated at 37 degrees C, 5% CO2 for 24 h. Following the initial 24-h incubation, the infected monolayers were scraped, syringe lysed with a 25 G needle, and filtered through a 5 μm syringe driven filter. Zoites were then pelleted and counted to set up subsequent infections.

### RNAseq of *H. hammondi* (HhAmer) and *T. gondii* (TgVEG) in control and bradyzoite induction conditions

HFFs were infected with 24 h zoites at an MOI of 0.5 for *T. gondii* and 8.5 for *H. hammondi*. After 48 h (3 days post-excystation; DPE), infected host cells were exposed to either bradyzoite induction conditions (pH 8.2, 1% serum, 0% CO$_2$[66]; or normal media and conditions. At 48 h, infected host cells were washed with PBS and RNA was harvested using the RNeasy Mini Kit 9 (Qiagen) and total RNA was processed for next generation sequencing as previously described[16]. CLC Genomics Workbench or bowtie2 was used to map all fastq reads to either the *T. gondii* genome or the *H. hammondi* genome as previously described[16]. Total gene counts were exported from CLC and filtered so that only genes with a total of 10 reads across all parasite-specific samples were used for differential expression analysis using DESeq2[67]. A total of 7182/8920 genes were analyzed for *T. gondii* and 6106/7266 for *H. hammondi*. When necessary genes were matched between species based on orthology information found on ToxoDB.org. Venn diagrams were created with BioVenn[68] and Venn Diagram Plotter software or manually. Heat maps were generated with TM4 MeV 4.9.0[69].

### Disruption of candidate genes in *T. gondii* VEG and *T. gondii* PRU and their complementation using CRISPR/CAS9

For each candidate gene specific guide RNAs (gRNAs) were designed using the E-CRISP design tool (*Toxoplasma gondii* genome, medium setting). The gene specific gRNAs were incorporated into a version of the the pSAG1::CAS9-U6::sgUPRT plasmid provided by the Sibley lab[70] that was engineered to so that the UPRT gRNA was replaced with an PseI and FseI restriction site[48] (pCRISPR_ENZ) using Q5 mutagenesis (NEB) and verified with Sanger sequencing. Linear repair constructs encoding either the HXGPRT and DHFR-TS selectable markers under the control of the DHFR 5' and 3' UTR were amplified from either the pGRA_HA_HPT[71] or pLIC-3HA-DHFR[72] using Platinum® *Taq* DNA Polymerase High Fidelity with High Fidelity PCR buffer and 50–150 ng of template per reaction according to the manufacturer's reaction conditions. Approximately $1 \times 10^7$ parasites were transfected with 25 μg of the NheI-HF linearized CRISPR plasmid and 2.5 μg of the appropriate linear repair cassette in 800 μL of Cytomix (10 mM KPO$_4$, 120 mM KCL, 5 mM MgCl$_2$, 25 mM HEPES, 2 mM EDTA) supplemented with 2 mM

ATP and 5 mM glutathione using a BTX electroporation system – Electro Cell Manipulator 600 (2.4 kV Set Charging Voltage, R3 Resistance timing). Transfection reactions were incubated at room temperature for 5 min before infecting confluent monolayers of HFFs. After ~24 h, the infected host cells were grown in selection media containing cDMEM supplemented with mycophenolic acid (MPA; 25 μg/mL) and xanthine (X; 50 μg/mL) for TgCEP or cDMEM with 1 μM pyrimethamine for TgVEG and TgPRUΔHXGPRT to select for parasites that incorporated the repair cassettes containing the selectable markers. After a stable population of parasites capable of growing in selection media was obtained, clonal population were obtained through serial dilution. Genomic DNA was used to verify disruption of the candidate gene locus with the selectable marker with PCR. Amplified products were visualized with 1% agarose gels run at 90 volts for 25–40 min and in some cases sequenced.

To complement PRUΔROCY1 knockout parasites the predicted cDNA was PCR amplified and either cloned directly into NsiI/NcoI digested pGRA-HA-HPT plasmid (leaving the GRA1 promoter from the plasmid intact) or fused to ~2000 bp of upstream sequence and then cloned into pGRA-HA-HPT digested with HindIII and NcoI (which removes the GRA1 promoter harbored in the plasmid). For C-terminal tagging the HA tag found within the pGRA-HA-HPT plasmid was used, while for N-terminal tagging a single HA tag was introduced after the start codon using PCR. These constructs were then validated by insert sequencing and then transfected into TgPRUΔROCY1, selected for drug resistance in MPA/X and cloned by limiting dilution.

### Alkaline pH-stress-induced tissue cyst formation assays

Confluent monolayers of HFFs grown on nitric acid-etched coverslips in 24-well plates were infected with *T. gondii* at an MOI ranging from 0.25 to 1 depending on the experiment. MOIs were the same for all tissue cyst formation assays performed on the same day. Parasites were grown for 48 h and then media was replaced with either control media (cDMEM pH 7.2) or bradyzoite induction media, pH 8.2,[66]. Parasites grown in control conditions were incubated at 37 degrees C, 5% CO$_2$ and parasites grown in alkaline pH stress conditions were incubate at 37 degrees C with ambient CO$_2$ (0.03%). All media were replaced every 24 h. At different time points post-exposure to BIC, infected coverslips were washed 2X with PBS, fixed with 4% paraformaldehyde in PBS, washed 2x with PBS, and stored in PBS at 4 degrees C until immunostaining was performed.

### Immunofluorescence assays

Fixed coverslips were blocked with 5% BSA in PBS with 0.15% Triton-X 100 for 1 h at room temperature. Coverslips infected with TgVEG WT, TgVEGΔ311100 (ROCY1), TgVEGΔ200385 (BFD1), TgVEGΔ207210, and TgCEPΔ221840 were stained with a primary polyclonal anti-MAF1b from mouse serum (1:1000 dilution[73];). After primary staining, the coverslips washed with PBS, stained with secondary goat anti-mouse Alexa-fluor 488 antibody and Rhodamine-labeled or biotinylated *Dolichos biflourus* Agglutin (1:250) and washed with PBS. For biotinylated DBA this was followed up by incubating in streptavidin coupled to either AlexaFluor 405 or AlexaFluor 647. Coverslips were then mounted using ProLong Diamond Antifade mountant or Vectashield (both with and without DAPI) and allowed to cure overnight at room temperature if necessary. Coverslips were blindly observed with a 100X objective on an Olympus IX83 microscope with cellSens software. The percentage of DBA-positive vacuoles was generally determined by counting vacuoles from 15 randomly chosen, parasite containing fields of view, although in some cases parasites were first identified to be expressing a transgene of interest prior to scoring. Images were exported as.tiff files and average fluorescence intensity per pixel was quantified using ImageJ software or pipelines developed in Knime[74].

### RNAseq of TgVEGΔROCY1 and TgVEGΔBFD1 in control and bradyzoite induction conditions in vitro

Confluent monolayers of HFFs grown in 24-well plates were infected with TgVEG WT, TgVEGΔROCY1, and TgVEGΔBFD1 parasites at an MOI of 0.5 and grown in control conditions for 48 h. After 48 h the media was replaced with either control media (cDMEM pH 7.2) or bradyzoite induction media[66]. Infected host cells grown in control conditions were incubated at 37 degrees C, 5% $CO_2$ and infected host cells grown in bradyzoite induction media were incubated at 37 degrees C with ambient $CO_2$ (0.03%). Control or bradyzoite induction media was replaced every 24 h. After 48 h of growth in bradyzoite induction conditions, all infected host cells were washed with PBS and total RNA was harvested using the RNeasy Mini Kit 9 (Qiagen). This RNA was then processed for strand-specific RNA sequencing as previously described[16]. Fastq files were first mapped to the human genome to remove host transcripts, (hg38) using CLC Genomics Workbench (default mapping settings for reverse strand sequencing, similarity fraction adjusted to 0.95) and a file of unmapped reads was generated for each sample. Unmapped reads were then mapped to the *T. gondii* genome (TgME49 v38) (default mapping settings for reverse strand sequencing). Total gene counts were exported from CLC and filtered so that a gene was only included for differential expression analysis using DESeq2[67] if there were a total of 30 or more reads across all samples. A total of 5970/ 8920 *T. gondii* genes were analyzed. Venn diagrams were created with BioVenn[68] and Venn Diagram Plotter software[75].

### Enhanced Crosslinking Immunoprecipitation (eCLIP)

[FLAG]ROCY1 parasites were inoculated onto confluent HFFs at an MOI of 1 and grown in bradyzoite-inducing media (RPMI, 50 mM HEPES pH=8.3, 5% FBS) for 48hrs. Infected monolayers were rinsed in PBS and UV crosslinked at 175mJ/cm$^2$ twice at 254 nm. Parasites were released from HFFs by syringe lysis with a 23 gauge needle and rinsed twice in PBS. Cell pellets were resuspended in 1 ml lysis buffer (50 mM Tris pH7.4, 100 mM NaCl, 1% NP40, 0.1% SDS, 0.5% sodium deoxycholate) and immunoprecipitated with anti-FLAG magnetic beads (Pierce A36797) for 6 h after taking a size-matched input (SMI) sample. IP and Isolation of IP-enriched and SMI samples and sequencing libraries from were generated in biological triplicates as per the eCLIP methodology[76,77]. Libraries were read by Illumina sequencing at 2x150nt and deposited at GSE211957.

### eCLIP data analysis

All data processing was conducted using software packages available through the Galaxy platform[78]. Sequencing adapters were removed with CutAdapt v4.0[79] and unique molecular identifiers were called with UMItools v1.1.2[80]. Reads were mapped against the *T. gondii* reference genome using RNA STAR v2.7.8a[81] with the settings −outFilterMismatchNoverLmax = 0.1, −outFilterScoreMinOverLread = 0.75, −outFilterMatchNminOverLread = 0.75, −outFilterMultimapNmax = 1, −alignIntronMax = 1. The reference genome (TGME49v58) and its annotation file were obtained from toxodb.org while the bfd1 gene model was obtained from[10]. Mapped reads were deduplicated with UMItools[80]. [FLAG]ROCY1-enriched regions were assessed by PureCLIP v1.0.4[46] using the setting −dm=50 and including a reference file of non-specific crosslink sites from the SMI samples. Peaks were annotated using Homer v4.11[82]. Enrichment of [FLAG]ROCY1 binding regions was obtained by generating heatmaps with deeptools2[83] and overlaying the metagene profile (5′UTRs, CDS, and 3′UTRs) of *Toxoplasma* transcripts as obtained in[84].

### Endogenous gene tagging

C-terminal endogenous tagging of ROCY1 was performed in TgPRUΔKu80ΔHXGPRT by targeting the 3′UTR of ROCY1 with a specific gRNA cloned into the pUniversal-CAS9 plasmid (Addgene #52694) and then co-transfected with a homology repair cassette amplified from the pLIC-HXGPRT plasmid (as outlined in[40,41]). Guide

RNA sequences and other oligonucleotides for this manipulation can be found in Supplementary Dataset 1. For N-terminal tagging of ROCY1 in TgPRUΔKu80ΔHXGPRT two gRNA target sites and their respective PAMs were identified using EuPaGDT (hosted by the University of Georgia[85];): one was 1632 bp upstream of the ROCY1 start codon and two others had PAMS that were 11 or 12 bp downstream of the predicted ROCY1 start codon. Oligos encoding these gRNAs were cloned into NcoI/PacI-cut pCAS9-GFP-T2A-HXGPRT (kindly provided by Drs. Joanna Young and Moritz Treeck, Francis Crick Institute, London[86];) using Gibson assembly and verified by sequencing. To generate the repair template for insertion of a FLAG tag in the N-terminus near the downstream gRNA cut site, a two step PCR was performed to amplify the full length ROCY1 promoter along with an HXGPRT resistance cassette, where each end of the cassette had a homology a reverse primer was designed with a homology arm to insert a single FLAG tag after the start codon in frame with residue 4 of the predicted ROCY1 protein (which begins with M E G A; see Supplementary Dataset 1 for primers from both steps and gRNA oligo sequences). Freshly harvested *T. gondii* PRUΔKu80ΔHXGPRT were transfected with 15 μg of this PCR product along with 3 μg of each gRNA plasmid (3 in total; one targeting -1.6 Kb upstream of the start codon and the others targeting the N-terminus of the coding region). Following selection with MPA/Xanthine clones were isolated by limiting dilution and screened for FLAG-tagged ROCY1 by induced cyst formation for up to 3 days in vitro and staining with M2 anti-FLAG monoclonal antibody or by harvesting protein and performing western blots.

### Generation of GFP luciferase parasites

TgVEG WT, TgVEGΔROCY1, and TgVEGΔBFD1 parasites were modified to harbor a GFP and click beetle luciferase expression cassette in the sinefungin resistance locus[87]. A gRNA targeting the SNR1 gene in *T. gondii* (TgVEG_290860) was designed using the E-CRISP design tool (*Toxoplasma gondii* genome, medium setting) and incorporated into the pCRISPR_ENZ plasmid using Q5 mutagenesis and verified with Sanger sequencing (pCRISPR_SNR1). Homology arms corresponding to the 20 bp sequencing flanking the Cas9 cut site were inserted to the pClickLUC-GFP plasmid (pClickLUC-GFP_SNR1-HA) using Q5 mutagenesis and verified with Sanger sequencing. Approximately $1 \times 10^7$ TgVEG WT, TgVEGΔROCY1, and TgVEGΔBFD1 parasites were transfected with 25 μg each of pCRISPR_SNR1 and pClickLUC-GFP_SNR1-HA using protocols described above. After ~24 h, the infected host cells were grown in selection media containing cDMEM and $3 \times 10^{-7}$ M sinefungin. After a stable population of sinefungin-resistant parasites were obtained, parasites were cloned using limiting dilution and screened for expression of GFP. To screen GFP-positive parasite clones for luciferase expression, parasites were scraped, syringe lysed with a 25- and 27-gauge needle, pelleted, and resuspended in PBS at a concentration of 500,000 parasites/mL, 250,000 parasites/mL, and 125,000 parasites/mL. For each concentration of parasites, 200 μL was added to each well of a black 96 well plate (100,000, 50,000, and 25,000 parasites per well) in triplicate and 50 μL of d-Luciferin potassium salt was added to each well. Parasites were incubated with d-Luciferin at room temperature for 10 min and the luciferase signal was measure using an IVIS Lumina II in vivo bioluminescence imaging system. Non-luciferase expressing parasites and PBS were used as negative controls. Following these manipulations parasite clones were validated as maintaining their ΔROCY1, ΔBFD1 or WT genotype using the PCR primers designed to characterize each knockout (Supplementary Dataset 1).

### Murine TgVEG WT-GFP-LUC, TgVEGΔROCY1-GFP-LUC, and TgVEGΔBFD1-GFP-LUC infections in CBA/J mice for cyst formation assays and RNAseq

For in vivo infections, 5-week-old CBA/J female mice (Jackson Laboratories) were infected with 250,000 TgVEG WT-GFP-LUC,

TgVEGΔROCY1-GFP-LUC, and TgVEGΔBFD1-GFP-LUC parasites in 200 μL of PBS via intraperitoneal injection. Mice were imaged ventrally using a 4 min exposure and large binning at 3 h post infection and on multiple days post-infection 10 min following intraperitoneal injection with 200 μL of d-Luciferin potassium salt as previously described[47]. After 21 d, 30 d or 9 weeks postinfection, mice were sacrificed and whole brains were removed (for the 30 d and 9 week infections a subset of these mice were used in immune suppression based reactivation experiments. See description of these methods below). A brain homogenate was prepared by passing half of the brain through a 100 μm cell strainer using 5 mL of PBS. An additional 20 mL of PBS was added to the homogenate and it was pelleted by spinning at 1000× g for 5 min and the pellet was resuspended in 1 mL of PBS. Livers were processed identically to brains. In some cases all or a portion of the brain or liver was put into culture on a confluent monolayer of HFFs. Flasks were incubated overnight after which time they were washed and allowed to incubate for up to 5 days before being syringe lysed and used to infect a new flask. If no parasites were observed the entire flask homogenate was passed to the next flask. A maximum of 5 passages were conducted to identify live parasites.

To quantify the number of parasite genomes in the brain homogenate, genomic DNA was extracted from 100 μL of brain homogenate using the GeneJET genomic DNA isolation kit according to manufacturer's instructions. qPCR was used to quantify the total number of genomes using primers targeting the *T. gondii* B1 gene and primers targeting mouse GAPDH as a control gene. All reactions were performed in duplicate using a QuantStudio 3 Real-Time PCR System. Samples were analyzed in 10 μL reactions consisting of 5 μL of 2X SYBR Green 2X master mix, 1 μL of 5 μM forward and 5 μM reverse primers, 2 μL of ddH2O, and 2 μL of genomic DNA. Cycling and melt curves were performed as previously[16]. To determine the total number of parasite genomes per brain, a standard curve of known parasite numbers was performed using *T. gondii* B1 primers. This DNA was also used to validate that each line maintained its ΔROCY1, ΔBFD1 or WT genotype using the PCR primers designed to characterize each knockout (Supplementary Dataset 1).

To quantify brain cyst burden, 100–300 μL of brain homogenate was fixed with 900–1,250 μL of 4% paraformaldehyde in PBS for 20 min. Fixed samples were pelleted at 5200x g for 5 min, washed with PBS and resuspended in 1 mL of PBS, and stored at 4 degrees C. Fixed brain homogenate was stained with Rhodamine-labeled *Dolichos bifluorus* Agglutinin (1:150) overnight at 4 °C with rotation. The following day, samples were pelleted (5200× g for 5 min) washed with 1 mL of PBS, pelleted, and resuspended in 1 mL of PBS. For each sample, 300 μL of stained homogenate was spread across 6 wells of a flat-bottomed 96 well plate and tissue cysts were blindly counted using an Olympus IX83 fluorescent microscope using the 10X objective. Cyst burdens were calculated by multiplying the total number of tissue cyst observed in the brain homogenate by the dilution factor. For immunohistochemistry brain portions were immediately placed into 10% neutral buffered formalin (Sigma) for at least 24 h and then processed for paraffin embedding, sectioning and staining at the University of Pittsburgh Histology Core, Department of Pediatrics. Parasites were stained using rabbit anti-GFP To quantify serum levels of IFNγ and IgG we bled mice from the submandibular vein into serum gel tubes (Sarstedt; product 41.1378.005) allowed to clot at room temperature for up to 30 min and then centrifuged at 800× g. For IFNγ we used the OptEIA™ Mouse IFN-γ (AN-18) ELISA kit (BD Biosciences) and for serum IgG we used the Mouse Toxoplasmosis (TOXO) Antibody (IgG) ELISA Kit, both according to the manufacturers directions.

For host gene expression analysis from infected mouse brains, brain homogenate aliquots were centrifuged at 800× g for 10 min, and RNA was isolated using the Qiagen RNeasy Mini kit with the protocol optimized for samples with high lipid content by resuspending the pellet in 1 mL of Qiazol (Qiagen) and freezing it at −80 °C prior to RNA isolation. Stranded mRNA libraries were constructed using the Illumina TruSeq Stranded mRNA kits or the NebNext Ultra II stranded mRNA kits and sequenced at the University of Pittsburgh core facility on a NextSeq 500 sequencer. Demultiplexed fastq files were mapped to the mouse genome GRCm38 using CLC Genomics Workbench (Qiagen) using default read alignment settings except for selecting "Reverse" on the strand-specific menu page. Data were exported as raw read counts and analyzed for differential transcript abundance using DESeq2[88], differentially regulated "HALLMARK" pathways using Gene Set Enrichment Analysis (GSEA[51];), and estimated immune cell proportions from bulk RNAseq data using Cibersort[53] using the LM22 signature matrix profile representing 22 immune cell types.

## Analysis of parasite gene expression during chronic infection in vivo

Female Balb/c mice aged 6–8 weeks were infected 100,000 tachyzoites of TgVEG WT-GFP-LUC or TgVEGΔROCY-GFP-LUC (*N* = 3 per strain) and monitored for infection using in vivo bioluminescence imaging. Mice were euthanized on D29 post-infection and the brain was harvested by dissection. To enrich for parasite RNA, one half of the brain was passed through a 100 μm cell strainer in PBS, washed in PBS by centrifugation at 1500xg for 15 min, resuspended in 3 mL of PBS and then passed sequentially through a 25 and 27 gauge needles attached to a 3 mL luer lock syringe. After washing in 40 mL of PBS and centrifuging as above, the pellet was resuspended in 5 mL of PBS and then homogenized further with 0.1 mm silica beads using a Sonibeast Junior (BioSpec Products; Bartlesville, OK) on the maximum speed setting for 10 seconds. Samples were centrifuged again, resuspended in 5 mL PBS and then passed through a sterile polycarbonate 5 μm filter. After two final washes in PBS the pellet was resuspended in Qiazol (Qiagen) and stored at −80 degrees C until RNA isolation. After isolating total RNA from each sample stranded RNA sequencing libraries were made using the NEB Next Ultra II Stranded Library preparation kit in conjunction with the NEBNext Poly(A) mRNA Magnetic Isolation Module (New England Biolabs). RNA libraries were sequenced using paired end chemistry on an Illlumina NextSeq 550, reads aligned to the *T. gondii* VEG genome (version 43; www.toxodb.org) using bowtie2 (default settings for paired sequencing, except we set -k = 5 and used the −very-sensitive-local toggle) and Samtools v1.9-66 for file conversion and sorting. We used featureCounts from the Subread[89] package to count reads per transcript using the command modified as follows: featureCounts -Q 10 -p -t gene -s 2 -g ID. The quality score cutoff was critical for accurate estimates of transcript abundance. Data were then analyzed using DESeq2 as above for host gene expression.

## Reactivation of chronic murine infections with TgVEG WT, TgVEGΔROCY1, and TgVEGΔBFD1

CBA/J female mice (Jackson Laboratories) were infected with 250,000 TgVEG WT-GFP-LUC, TgVEGΔROCY1-GFP-LUC, and TgVEGΔBFD1-GFP-LUC parasites in 200 μL of PBS via intraperitoneal injection and imaged at multiple times post-infection using in vivo bioluminescence imaging as described above. At the indicated time post infection (30 days, 9 weeks, or 5 months), and when there was no detectable parasite-derived luminescence in any tissue, the reactivation group was given dexamethasone (20 mg/L) in their drinking water *ad libitum* while the control group continued to be given normal drinking water. Mice were imaged (as described above) prior to dexamethasone treatment (0 days post dexamethasone) and on multiple days post-dexamethasone treatment, and weight loss, parasite-derived luminescence and other signs of infection were used to determine parasite burden and the extent of recrudescent-disease driven morbidity. Data were collected either until most of the mice in at least one treatment group had become morbid due to infection and reactivation (30 day and 9 week reactivation), or until clear recrudescent bioluminescence could be detected (5 month reactivation).

## Infections in mice for immune cell infiltration studies and flow cytometry

For flow cytometric analyses, C57BL/6 mice were infected with 200 TgVEG WT-GFP-LUC or TgVEGΔ*ROCY1*-GFP-LUC parasites in 200 μL of PBS via intraperitoneal injection. At day 28 post-infection, mice were transcardially perfused with 50 mL of 1X PBS to remove non-adherent blood cells. Brains were harvested and digested using Dispase II (Roche Applied Science) diluted in Hepes-buffered saline. To remove myelin, 35% and 75% percoll (GE Healthcare) gradients were used. To block non-specific binding of antibodies to immune cells, the isolated cells were resuspended in 10% TrueStain FcX Buffer (BioLegend) in staining buffer (3% fetal bovine serum in 1X PBS). Cells were surface-stained with fluorescent dye-conjugated antibodies diluted in staining buffer (Biolegend: Ly6G:BV510 Cat#127633, CD11b:BV605 Cat#101257, CD45:BV785 Cat#103149, Ly6C:PerCp-Cy5.5 Cat#128017, CD3:APC-Cy7 Cat#100222). Cells were then resuspended in 1X PBS and run on the Novocyte flow cytometer (Agilent). Flow cytometry data were analyzed and graphically represented utilizing FlowJo software (Treestar), and the gating strategy is outlined in Fig. S6.

## Reporting summary

Further information on research design is available in the Nature Portfolio Reporting Summary linked to this article.

## Data availability

All next generation sequencing data (in the form of Fastq files) have been deposited in the Gene Expression Omnibus (GEO) database at https://www.ncbi.nlm.nih.gov/geo/query/acc.cgi?acc=GSE211957 for the eCLIP data, https://www.ncbi.nlm.nih.gov/geo/query/acc.cgi?acc=GSE218921 for pH responses of *T. gondii* and *H. hammondi*, and https://www.ncbi.nlm.nih.gov/geo/query/acc.cgi?acc=GSE218920 for RNAseq analysis of mouse brains after infection with wild type and mutant *T. gondii*. Source data for graphs are provided in the Source Data File. Source data are provided with this paper.

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

## Acknowledgements

This work was funded by grant NIH F31AI140529 to S.L.S.B, NIH T32GM133353 and F31AI167594 to L.F.C., NIH T32GM133353 to H.S.A., NIH F31NS118865 to S.B.O., NIH R01AI116855 to J.P.B., NIH R21AI167662 and NIH R01AI124723 to W.J.S. and NIH R01AI120846 to M.B.L.

## Author contributions

S.S.B.[1*] conceived, designed, executed and analyzed 2/3rds of the experiments and wrote 2/3rds of the manuscript. S.M.R. conceived, designed, executed and analyzed 1/3rd of the experiments and wrote 1/3rd of the manuscript. M.H. designed, performed and analyzed eCLIP experiments and wrote the methods and created initial figures for that experiment. S.O. performed the flow cytometry experiments and generated figures for them. M.D.M. performed and analyzed experiments in Fig. 8. K.G.S. performed and analyzed experiments in Figs. 3 and S3. L.F.C. generated the data in Fig. S7 concerning host cell gene expression. H.S.A. generated transgenic parasites for the expeirments in Figs. 2 and 3. B.M.D.G. unpublished data that guided in vivo experiments. M.L. helped to conceive design and interpret the flow cytometry experiments. W.J.S.Jr helped to conceive, design and interpret the eCLIP experimnets. J.P.B. helped to conceive of, design and interpret the experiments performed by S.S.B. and S.M.R. and other authors with the affiliation at the University of Pittsburgh. J.P.B. also revised and edited the manuscript throughout the review process and altered and updated figures as new data were provided.

## Competing interests

The authors declare no competing interests.

## Ethics

All procedures involving animals were approved by the local IACUC at either the University of Pittsburgh or the University of California Irvine. Laboratory safety and recombinant DNA procedures were approved by the local Environmental Health and Safety and Institutional Biosafety Committees at the University of Pittsburgh, Indiana University School of Medicine, and University of California Irvine. For animal experiments female mice were used in all studies as this is the most well-characterized animal model for *T. gondii* infection and our goal was to examine cyst formation of our mutants rather than directly assess the impact on the host. It would be, in this case, ethically questionable to perform all experiments in both males and females as this would require additional pilot studies in males and double the number of animals used in our experiments unnecessarily.
