## [Peer Review File · Nature Communications]

A transcriptional network required for bradyzoite development in *Toxoplasma gondii* is dispensable for recrudescence diseaseEditorial Note: This manuscript has been previously reviewed at another journal that is not operating a transparent peer review scheme. This document only contains reviewer comments and rebuttal letters for versions considered at Nature Communications.

Reviewers' Comments:

Reviewer #1:

Remarks to the Author:

This is a manuscript from Sokol Borrelli et al. entitled "A transcriptional network required for cyst formation in the opportunistic pathogen *Toxoplasma gondii* is dispensable for recrudescence disease". The authors did address some of the concerns that were raised by both reviewers. However, they did not alleviate some of the concerns.

Major points:

- 1- Both reviewers pointed out earlier that there is no direct evidence that ROCY1 KO strain lacks a cyst wall. Although some efforts have been made to attenuate this claim, it remains a central point that the authors are trying to force even though it is not supported by data. In some parts of the text (such as line 55) this affirmation still stands while in others the authors call the ROCY1 KO parasites cyst wall-defective (line 450) without direct evidence (other than the indirect gavage experiment). Overall, the fact ROCY1 KO strain lacks an effective cyst wall is only based on DBA staining. The fact that this is brought forward throughout the text as a paradigm-shifting piece of evidence is obscuring the rest of the nicely controlled and well-crafted work. This is a critical aspect that has been brought forward from the beginning of the review process and that is still a major issue in the current manuscript.
- 2- Since the BFD2 study has been published (Licon et al., 2023), it would make sense that the authors compare their dataset with what has been published. This is particularly important for the e-CLIP dataset that presents a large number of targets and that could be compared to the RIP-Seq data.
- 3- The authors provide an important piece of evidence in Figure 7F showing that the brain of mice infected by both BFD1 and ROCY1 KO strains are filled with GFP-positive (yet DBA-negative) parasite vacuoles. It seems fairly easy to address the presence of a cyst wall by electron microscopy as suggested previously by both reviewers.

Licon, M.H., Giuliano, C.J., Chan, A.W., Chakladar, S., Eberhard, J.N., Shallberg, L.A., Chandrasekaran, S., Waldman, B.S., Koshy, A.A., Hunter, C.A., Lourido, S., 2023. A positive feedback loop controls *Toxoplasma* chronic differentiation. *Nat. Microbiol.* 8, 889–904.
<https://doi.org/10.1038/s41564-023-01358-2>